# Elevated carbon monoxide observations over the North Pacific upper troposphere (July 2012–February 2013)

**Kuo-Ying Wang** [1]*, **Philippe Nedelec** [2], **Valerie Thouret** [2], **Hannah Clark** [2], **Andreas Wahner** [3], **Andreas Petzold** [3]

1 Department of Atmospheric Sciences, National Central University, Chung-Li, Taiwan, 2 Laboratoire d'Aerologie, Centre National de la Recherche Scientifique, Observatoire Midi-Pyrenees, Toulouse, France, 3 Forschungszentrum Julich GmbH, Institut fur Energie und Klimaforschung, IEK-8 Troposphare, Julich, Germany

* kuoying@mail.atm.ncu.edu.tw

**Data availability statement:** The data underlying the results presented in the study are available from IAGOS (https://www.iagos.org).

## Abstract

A novel use of routine, commercial, and regular in-service passenger airplane in collecting chemistry data over the North Pacific upper troposphere for climate research was started in July 2012. The European In-Service Aircraft for a Global Observing System (IAGOS) Package-1 was installed aboard a commercial Airbus A340-300 (B18806) operated by China Airlines (CAL). In this work we present carbon monoxide (CO) collected during the period from July 2012 to February 2013. Our findings, when compared with NASA aircraft missions in 1991, 1994, and 2001, indicate increase in anthropogenic CO pollution in the North Pacific upper troposphere from 1991 to 2012. Elevated CO concentrations (>160 ppbv) were prominent in the summer months of July and August, followed by September, while concentrations were lower (<100 ppbv) from October 2012 to February 2013. The results underscore effective vertical transport of ground-level anthropogenic pollutants to the upper troposphere. HYSPLIT model calculations affirm that Asian industrial regions at ground level are the primary sources of CO pollution in the downwind areas of the North Pacific upper troposphere. This study enhances our understanding of temporal variations and sources of CO pollution in this specific region, utilizing routine in-situ measurements on commercial aircraft to contribute valuable insights to air quality and pollution transport dynamics.

## Introduction

Air pollutants are common byproducts of modern societies heavily reliant on fossil fuels [1]. These pollutants have diverse impacts on human health [2], surface biological processes [3], hydrological cycles [4], weather [5], and climate patterns [6]. Over the past 50 years, a comprehensive network of air quality monitoring stations and remote ground-level monitoring sites has continuously tracked trends in air pollution. When coupled with models, these measurements can unveil the sources of air pollution emanating from distant regions [7,8].

**Funding:** National Science and Technology Council, Taiwan. CNRS, Frace. Forschungszentrum Julich GmbH, Germany. The funders had no role in study design, data collection and analysis, decision to publish, or preparation of the manuscript.

The dispersion of surface pollutants to remote regions due to atmospheric dynamics poses challenges in obtaining measurements in the boundary layer and free troposphere. While ample data exist for surface regions, the subsequent evolution of air pollutants away from emission areas remains less understood, given the lack of tools for in-situ monitoring above the surface.

In-situ measurements offer valuable and precise data for comprehending the causes of air pollution issues. For instance, the NASA Transport and Atmospheric Chemistry near the Equator-Atlantic (TRACE-A) missions in August 1992 revealed that elevated ozone concentrations over the South Atlantic Ocean resulted from biomass burning emissions in Africa and South America [9].

Satellites routinely measure air pollution by detecting outgoing radiance emitted by target species, while lidar measurements on ships and on the ground gauge pollution through the absorption and reflection of specific light sources [10,11]. Various satellite instruments, such as the atmospheric infrared sounder (AIRS) on NASA Aqua satellite and the Scanning Imaging Absorption Spectrometer for Atmospheric Chartography (SCIAMARCHY) on ESA Envisat-1, provide measurements of CO and CO2. In-situ measurements are crucial for validating satellite data [12] and verifying emissions inventories [13].

Considering the extensive anthropogenic emissions from Asian regions [14], obtaining routine measurements in the Pacific upper troposphere is crucial for understanding the spatial and temporal extent of CO concentrations. In this work, we present continuous in-situ measurements of CO and ozone ($O_3$) conducted during routine commercial flights, extending the observational record established by the NASA Pacific Exploratory Mission-West (PEM-WEST) A, PEM-WEST B, and the Transport and Chemical Evolution over the Pacific (TRACE-P) experiments. Unlike previous intensive but short-term missions, our measurements spanned 8 months from July 2012 to February 2013, allowing us to assess pollutant distribution in regions distant from emission sources.

The detailed analysis of IAGOS Pacific flights and the spatial distribution of CO measurements over the North Pacific upper troposphere, between $120°E$ (eastern Eurasia continent and the beginning of the North Pacific atmosphere) and $120°W$ (western edge of North America, end of the North Pacific atmosphere), during an 8-month period (July 2012 to February 2013) has not been previously reported. These measurements taken during this 8-month period over such a wide-spread of the North Pacific upper troposphere can only be made possible by the long-hull flights using A340-300 airplane to across North Pacific. Hence, these in-situ flight measurements are rare to characterize long-range transport of CO from Asia to the downwind of the North Pacific upper troposphere.

Barret et al. [15] used IAGOS data to validate satellite (IASI-A, IASI-B, and IASI-C; Infrared Atmospheric Sounding Interferometer) CO measurements, covering the period 2008 to 2019. They presented results from Taipei ($25.09°N$, $121.24°E$) and Nagoya ($34.85°N$, $136.81°E$), located over the Northwestern Pacific, to verify the IAGOS measurements. Barret et al. [15] showed time-series results of CO in the troposphere between the surface–600 hPa layer and the 600 hPa–200 hPa layer over Taipei City for two periods: July 2012–February 2013 and 2015–mid 2018. Additional data were available from container measurements made by CARIBIC.

Lebourgeois et al. [16] used IAGOS measurements from 2002 to 2019, the FLEXPART trajectory model, and surface CO emissions inventories to attribute the ground-level sources of elevated CO measured by IAGOS on a global scale. Elevated CO was linked to surface emissions (anthropogenic, biomass burning, or a combination of both) by analyzing the 20-day backtrajectory from each elevated CO measurement. In the East Asia upper troposphere—defined as $90°E$ to $150°E$, and $15°N$ to $45°N$- —elevated CO levels (175 ppb) occurred during

the summer months, compared to 130 ppb in winter. During summer (June, July, August), about 95% of elevated CO originated from anthropogenic emissions, with the remainder from biomass burning and mixed sources. In winter (December, January, February), approximately 90% of elevated CO was attributed to anthropogenic sources, with the remainder from biomass burning and mixed emissions.

Cohen et al. [17] discussed the climatology and long-term evolution of ozone and carbon monoxide in the upper troposphere–lower stratosphere (UTLS) at northern midlatitudes, as observed by IAGOS from 1995 to 2013 on a global scale. In the East Asia region, defined as $105°E$ - $145°E$, and $30°N$ - $50°N$, elevated CO levels were found in the upper troposphere during late spring (May) and summer (June and July), while the lower stratosphere exhibited elevated CO levels during the summer (June, July, August). Cohen et al. [17] also reported negative CO trends (5th percentile, mean value, and 95th percentile) over the $105°E$ - $145°E$, and $30°N$ - $50°N$ area of the East Asia upper troposphere. It is noteworthy that IAGOS Pacific measurements commenced in July 2012 using China Airlines' in-service A340-300 aircraft.

Tsivlidou et al. [18] employed FLEXPART 20-day backtrajectories based on elevated CO measured by IAGOS over China and the tropical West Pacific ($25°N$ - $25°S$) during 2002–2020 to attribute the sources of elevated CO. The methodology used was similar to that of Lebourgeois et al. (2024). Again, IAGOS Pacific measurements were initiated in July 2012.

The aforementioned studies utilized extensive data periods (2008–2019, 2002–2019, 1995–2013, 2002–2020) from IAGOS measurements, covering areas primarily over the Atlantic, Western Europe, North America, South America, and Africa, as well as some flights from Europe to Japan, China, and Korea. In contrast, the IAGOS Pacific measurements, started in July 2012, were conducted over the open ocean of the North Pacific ($120°E$ - $120°W$) for flights from East Asia to North America.

Previous studies using IAGOS data have provided a broad, seasonal view of CO characteristics over the western edge of the Eurasian continent and at the intersection of Eurasia and the northwestern Pacific. In this work, we complement these long-term and seasonal studies by providing a detailed, zoomed-in analysis on a day-to-day and month-to-month basis for the first 8 months of IAGOS Pacific flights across the open ocean of the North Pacific upper troposphere. This detailed analysis captures episodic events that are missed in seasonal data and reveals transient CO peaks critical for understanding episodic pollution transport. We focus on July 2012–February 2013, marking the initiation of IAGOS Pacific flights, which provides a baseline for subsequent trends, supported by a sufficient data density for detailed analysis. Future work may explore later years to assess temporal evolution.

We present the distribution of elevated CO over the North Pacific open ocean upper troposphere and conduct backtrajectory calculations to identify the sources of both elevated and low CO. For the lower troposphere, we analyze a total of 520 CO profiles over Taipei (at a vertical resolution of 50 m), performing profile-by-profile backtrajectory analysis for both elevated and low CO measurements. The comparisons reveal distinct sources of air masses arriving in Taipei, complementing the upper tropospheric focus on sources of elevated versus low CO.

## Data and methods

### The PGGM project

The Pacific Greenhouse Gases Measurement (PGGM) project commenced in June 2009, utilizing a fleet of 9 in-service commercial container cargo ships from Evergreen Marine Corporation (EMC). Its primary objective was to conduct long-term in-situ measurements of

atmospheric carbon dioxide ($CO_2$) concentrations over the Pacific and North American service routes. Additionally, measurements were extended over the Pacific-Arab Peninsula-Europe service routes, traversing the Indian Ocean, the Red Sea, the Mediterranean Sea, and the northeast Atlantic Ocean [19,20].

Since July 2012, the PGGM project has engaged in collaboration with the European In-service Aircraft for a Global Observing System (IAGOS) consortium [21]. This collaboration involves the utilization of instruments developed by the IAGOS consortium for in-situ measurements of air pollutants, greenhouse gases, and cloud particles over Pacific regions. The PGGM project specifically employs an in-service routine commercial passenger aircraft, the Airbus A340-300 (designated code B18806), operated by Taiwan China Airlines (CAL). Notably, the IAGOS measurements installed on the CAL B18806 mark the inaugural routine in-situ measurements of carbon monoxide (CO), ozone ($O_3$), water vapor ($H_2O$), and cloud particles over the Pacific atmosphere [22].

It is worth mentioning that the Japanese CONTRAIL project has been employing a fleet of five Boeing 747 and 777 aircraft for routine in-situ measurements of greenhouse gases, namely carbon dioxide (CO2) and methane ($CH_4$), over the Pacific regions since 2005 [23].

## The IAGOS package 1 instruments

The IAGOS project [21] is a continuation of the highly successful MOZAIC project [24], initiated in 1994. Over the past two decades (1994-2014), MOZAIC has developed instruments installed on five Airbus A340-300 in-service commercial passenger aircraft for routine in-situ measurements of $O_3$, CO, $H_2O$, and nitrogen oxides (NOx). Demonstrating the feasibility of persistent and sustainable measurements with routine commercial passenger aircraft, the MOZAIC project has contributed valuable data to climate research [25].

The IAGOS Package 1 instrument was installed on a China Airlines A340 aircraft (B18806) in June 2012 [27]. This marks the inaugural routine and long-term in-situ measurements of CO, $O_3$, $H_2O$, and cloud particles over the Pacific regions by a commercial passenger aircraft [26]. The details of the IAGOS Package 1 are extensively described by *Nedelec et al.* [27], and the recorded data are output at a time resolution of every 4 minutes during the flight period.

The CO instrument is based on a standard Model 48 Trace Level (Thermo Scientific), akin to the one employed in the MOZAIC system [27]. Two decades of MOZAIC measurements have affirmed the reliability of this instrument for continuous operation over 6-12 months aboard commercial passenger aircraft.

Operating on the infra-red (IR) absorption method, the CO instrument utilizes an IR source at 4.6 $\mu m$. Filtering out $O_3$ and water vapor before measurements prevents interference with the CO signal. With an integration time of 30s using the Beer-Lambert Law, the instrument boasts a precision of $\pm 5$ ppbv and a detection limit of 10 ppbv. Considering the A340-300's maximum cruising speed of 250 m $s^{-1}$, this yields a horizontal resolution of about 7.5 km, and 450 m during ascent and descent.

To maintain the instrument's zero CO level unaffected by temperature variations, periodic CO zeroing of sampled air is performed with a Sofnocat filter. This zeroing occurs continuously on the ground, every 10 minutes during take-off and landing, and every 20 minutes during the cruise phase of the flight. The CO instrument undergoes removal from the aircraft every 6 months for laboratory calibration using NIST reference CO cylinders (CO in $N_2$, at 500 ppmv) and a dilution system. The CO calibrations cover levels at 0, 250, 500, 750, 1000, and 1500 ppbv to verify the instrument's linearity within 2%. The dilution system, calibrated

annually by the French LNE for flow meter calibration, ensures accuracy. Further details on the CO system and the IAGOS Package operation methods are available in Nedelec et al. [27].

## Model analysis of observational data

Utilizing measurements obtained during the cruising stage of flight, we conducted back trajectory calculations to investigate the source regions of cases exhibiting high and low carbon monoxide (CO) concentrations. The National Oceanic and Atmospheric Administration (NOAA) Hybrid Single Particle Lagrangian Integrated Trajectory model (HYSPLIT) was employed for this analysis [28].

For instances with elevated CO concentrations, we conducted back trajectory calculations during specific periods identified from the flight data, focusing on intervals when CO levels surpassed 100 ppbv in the upper troposphere (9-12 km). Out of the total 520 flights conducted throughout the study period (July 2012-February 2013), 394 cases were chosen. To ensure a uniformly distributed sampling across time, no more than four cases per flight were included. It's worth noting that there were instances where an entire flight recorded minimal CO levels, resulting in no cases being selected.

Conversely, for the low CO cases, back trajectory calculations were conducted for selected periods characterized by CO concentrations below 45 ppbv in the upper troposphere. A total of 208 cases were chosen for analysis. Each case underwent a 10-day back trajectory calculation, providing longitudes, latitudes, and altitudes at one-hour intervals. These data were then binned into a three-dimensional grid resolution with 1-degree longitude-latitude and 1-km vertical resolutions for subsequent analysis.

## The NASA aircraft missions over the Pacific regions

In addition to the commencement of PGGM/IAGOS measurements of carbon monoxide (CO) over the Pacific regions in 2012, prior measurements of CO were undertaken during NASA's PEM-WEST A, PEM-WEST B, and TRACE-P missions. The PEM-WEST A missions comprised 18 flights conducted in September-October 1991 [29]. Following that, the PEM-WEST B missions involved 16 flights conducted in February-March 1994 [30]. The TRACE-P missions further contributed to this dataset, encompassing 17 flights with the NASA DC-8 and 21 flights with a P-3B in February-April 2001 [31].

## Results

### The flight routes of CAL A340-300 B18806 aircraft

**Original flight routes.** The flight paths for B18806, spanning from July 2, 2012, to February 15, 2013, encompass a geographic expanse covering the North Pacific, East Asia, the tropical western North Pacific, and two distinct flight routes concentrated over the Euro-Asian continent (refer to Fig 1a). These flights traverse altitudes from the surface to 12 kilometers, spanning latitudes between $5°S$ and $70°N$ (see Fig 1b), and longitudes from $0°E$ to $120°W$ (see Fig 1c). Measurement profiles were obtained during both the ascending and descending phases of each flight, with a focus on regions between $100°E$ and $140°E$, $0° – 10°E$, and in proximity to $120°W$. A total of 520 flights were conducted, with daily routes varying from one (for long-haul flights) to five per day (for short-range flights, as depicted in Fig 2).

**Gridded data density from flight routes.** In Fig 3a, the spatial distribution of all measurements is presented, binned to a horizontal resolution of 1 x 1 degree in longitude-latitude. The areas with the highest data density, exceeding 2000 data points per grid cell, are concentrated in the tropical western North Pacific and the western North Pacific, attributed to

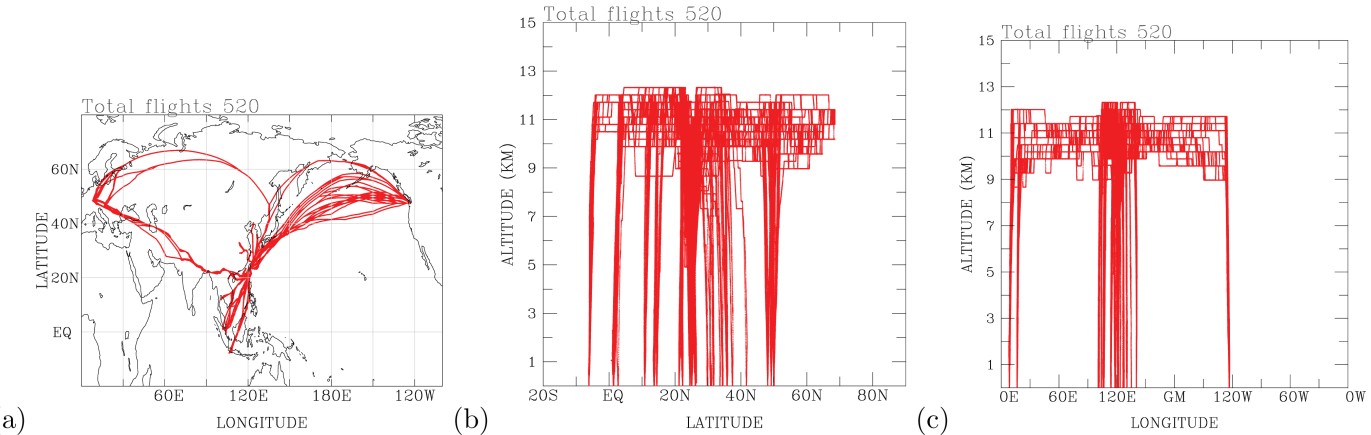

**Fig 1. The flight routes of the PGGM/IAGOS CAL B18806 A340-300 aircraft.** (a)longitude-latitude; (b) latitude-altitude; and (c) longitude-altitude plots.

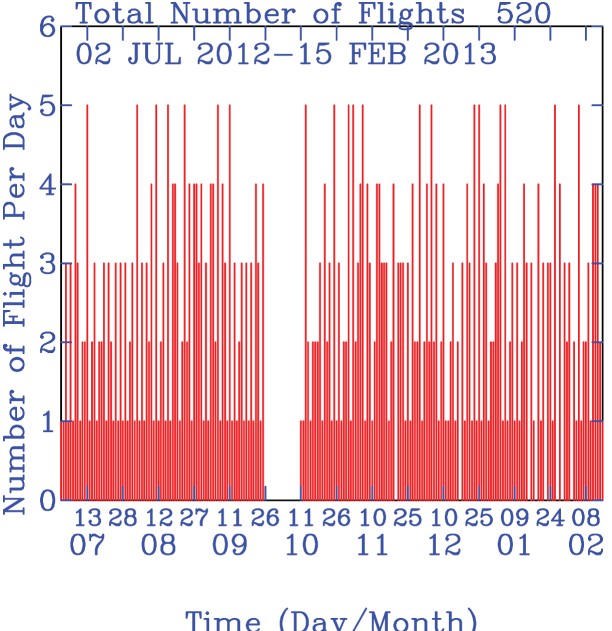

**Fig 2. Time-series plots of daily flight numbers.**

the frequent daily flights in this region. Additionally, elevated data densities are observed in the Euro-Asia region. The flight routes over the northern North Pacific exhibit diverse data densities, ranging between 100 and 2000 data points per 1-degree grid cell.

Vertically, Fig 3b illustrates that the highest data densities, ranging from 7000 to 15000 data points per 1-degree grid cell, are situated between 10-12 km altitude, spanning latitudes from the equator to 70N. In terms of longitude (see Fig 3c), this high data density region is positioned between $110°E$ and $130°E$. Over the northern North Pacific flight routes, data densities vary between 700 and 5000 data points per 1-degree grid cell, specifically in the longitudes spanning from $130°E$ to $120°W$ and at altitudes of 10-12 km. Notably, the data densities over the 10-12 km altitude of the Euro-Asia longitudes also fall within the range of 700-5000

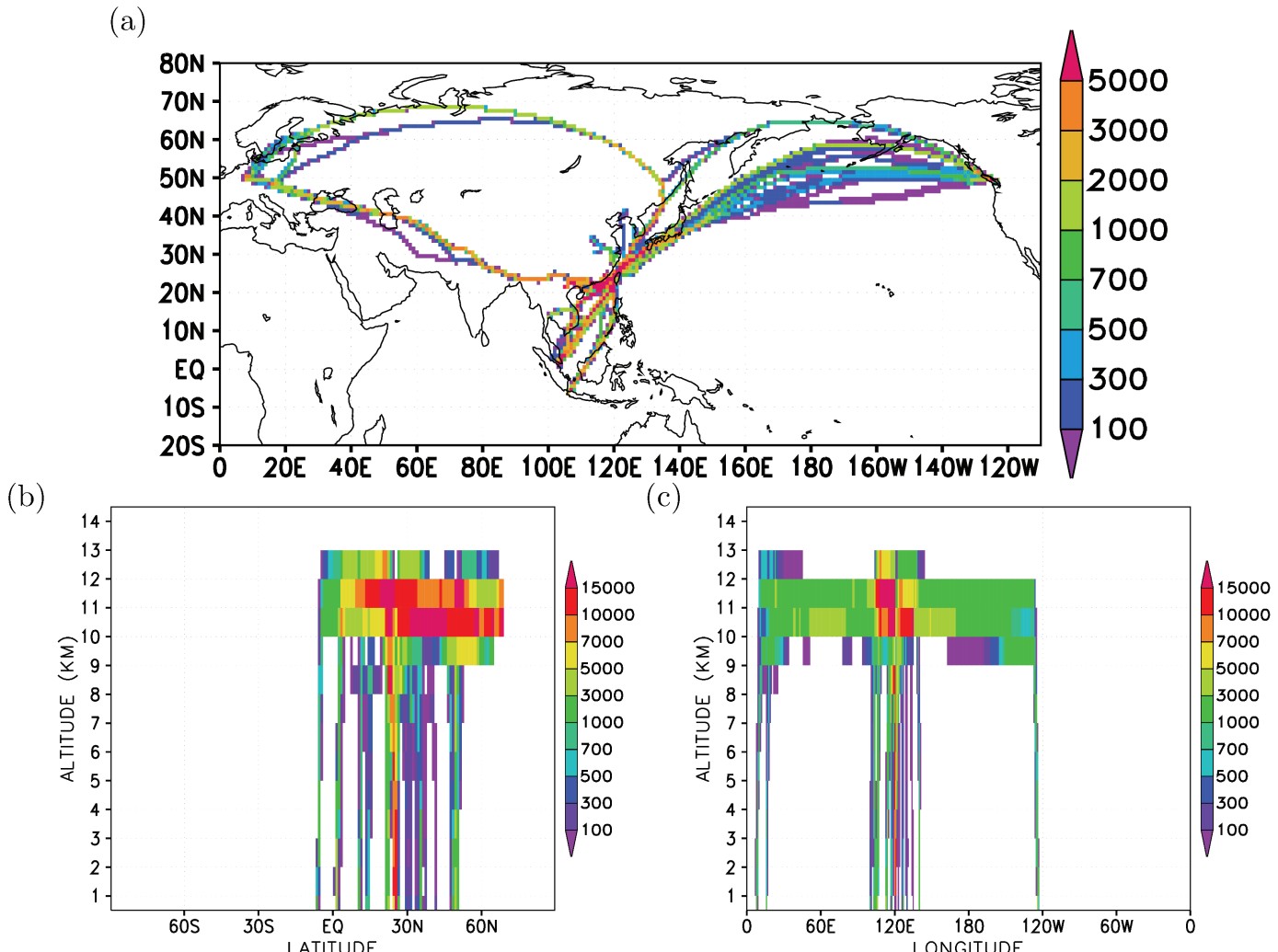

**Fig 3. Gridded flight routes for B18806.** (a) 1-degree longitude-latitude; (b) 1-degree latitude by 1-km altitude; and (c) 1-degree longitude by 1-km altitude distributions.

data points per 1-degree grid cell. This symmetry demonstrates the balanced operation of the commercial aircraft concerning Taipei ($120°E$, as depicted in Fig 3c).

### PGGM/IAGOS CO measurements

**Characteristics of a typical flight measurement.** In Fig 4a, we present the characteristics of a typical measurement taken from Taipei to Vancouver during the period from 15:00 UT on July 25 to 02:30 UT on July 26, 2012. The entire 11.5-hour flight over the northwestern Pacific route is illustrated in the upper panel, while the lower panel displays the flight trajectory colored with carbon monoxide (CO) concentrations. The upper panel also showcases time-series measurements of CO and ozone ($O_3$) concerning altitude during the flight. Elevated CO levels surpassing 150 ppbv at altitudes above 9.6 km were observed during specific periods: 16:30-17:30, 19:30, and 21:30-23:50, respectively. Conversely, very low CO levels (below 50 ppbv) were noted during 00:20-01:00 when high $O_3$ concentrations reaching 440

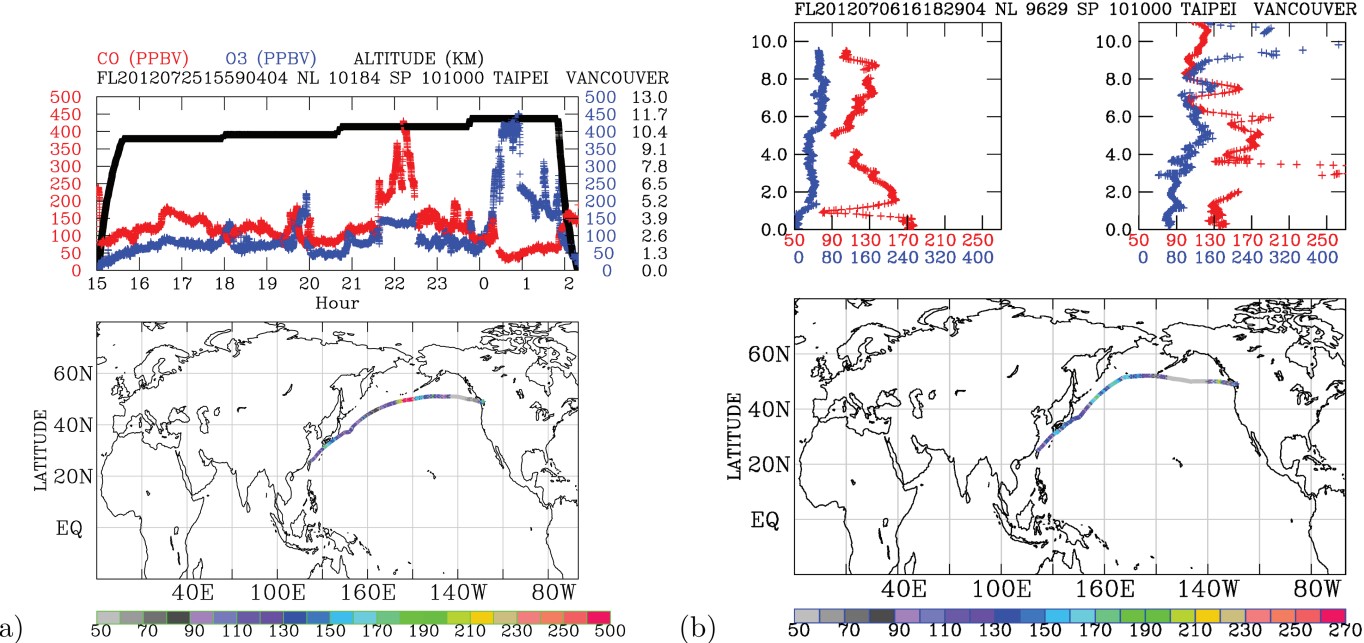

**Fig 4. Measurement taken from Taipei to Vancouver.** (a) Concentrations of CO (red) and $O_3$ (blue), along with flight altitude (black), during an IAGOS B18806 flight on 25 July 2012 (upper panel); horizontal distribution of CO concentrations across the North Pacific (lower panel; CO concentrations shown in color bar, units: ppbv). (b) CO and $O_3$ concentrations during the ascending (upper left panel) and descending (upper right panel) stages of a flight across the North Pacific on 6 July 2012; horizontal distribution of CO concentrations across the North Pacific (lower panel; CO concentrations shown in color bar, units: ppbv).

ppbv were measured, indicating the detection of stratospheric air by the aircraft. This CO-$O_3$ anti-correlation pattern also briefly occurred just before 20:00 hours, with simultaneous measurements of 200 ppbv of $O_3$ and 40 ppbv of CO.

During a one-hour period from 21:30-22:30, elevated CO concentrations ranging from 250-270 ppbv were observed at altitudes close to 10.8 km, alongside $O_3$ concentrations of 140 ppbv. This simultaneous elevation of CO and $O_3$ points to the long-range transport of ground-level pollutants into the upper troposphere.

Fig 4b illustrates two typical profiles of CO measured during takeoff from Taipei (top-left panel) and descent to Vancouver (top-right panel) on July 6, 2012. The ascending stage depicted CO concentrations reaching 170 ppbv near the ground, gradually decreasing to 90 ppbv at 5-km altitude, followed by a gradual increase to 130 ppbv at 9-km altitude, and a subsequent drop to about 100 ppbv at 10-km altitude. A sharp decrease in CO to about 80 ppbv occurred at 1-km altitude. Below 1-km altitude, a vertical profile exhibited nearly constant $O_3$ concentration of 50 ppbv and CO concentration of 170 ppbv. This indicates that the height of the night-time boundary layer (the aircraft took off at 16:18 UT, equivalent to 00:18 local time) is approximately 1 km. The vertical $O_3$ profiles reveal approximately four-layered structures: below 1 km (50 ppbv $O_3$ and 170 ppbv CO), 1-5 km (70 ppbv $O_3$ with CO decreasing with height), 5-8 km (close to 90 ppbv $O_3$ with CO increasing with height), and above 8-km altitudes (80 ppbv $O_3$ with CO decreasing with height). These intricate vertical structures, uncovered by profile measurements, are unique to PGGM/IAGOS flights and represent the routine profile measurements of combined CO and $O_3$ over eastern Asia.

During the descent into Vancouver, around 100 ppbv of CO and 400 ppbv of $O_3$ were measured in the upper troposphere (close to 10-km altitude), indicating the influence of stratospheric air. As the aircraft descended, elevated CO of 170 ppbv was measured at about 9-km, 4-6 km, and 3-km altitudes, respectively. The ground-level CO was approximately 130 ppbv. Notably, an area exhibited exceptionally elevated CO concentrations exceeding 250 ppbv between 2.5-3 km altitudes, signifying the intrusion of polluted air in this thin layer.

Fig 4 underscores the significance of the unique data obtained through PGGM/IAGOS measurements. Elevated CO levels were observed at various temporal and spatial locations during this flight. As CO serves as a reliable tracer for ground-level pollutants, the presence of elevated CO in the upper troposphere indicates the effective vertical transport of pollutants in the troposphere [32]. These in-situ CO measurements are crucial for validating model transport processes and satellite remote sensing of CO in the troposphere. The profiles of CO and $O_3$ reveal intricate vertical layers that are rarely measured over the Pacific regions.

**Profiles of CO distribution.** In Fig 5a, we present the complete dataset of carbon monoxide (CO) concentrations from 520 flights. Below 3 km altitude, CO concentrations decrease with height, ranging from 900 ppbv close to the surface to about 200 ppbv at 3 km altitude. Between 3 km and 9 km, most CO concentrations remain lower than 200 ppbv. From 9 km to about 12.5 km, elevated CO concentrations exceeding 200 ppbv were measured. This reveals three distinct vertical layers: below 3 km altitudes (CO decreases with height), between 3 km and 9 km altitudes (mostly constant CO concentrations), and between 9 km and 12.5 km altitudes (elevated CO higher than 200 ppbv). Notably, the lowest CO concentrations below 9 km are around 60 ppbv, while the lowest CO concentrations above 9 km are 30 ppbv. This pattern aligns with the surface being the primary source of CO.

When measurements are segregated by region, we can compare CO concentrations across different areas. In Fig 5b, profiles of CO measurements over developed countries in Europe (longitudes $0°E$ to $40°E$) reveal that CO concentrations are highest close to the surface, with levels less than 300 ppbv. CO concentrations decrease with height, ranging mostly between 100-200 ppbv at the surface and diminishing to 30-130 ppbv at 12 km altitude. Over the Euro-Asia content (longitudinal bands $40°E$ – $90°E$; Fig 5c), CO concentrations in the upper troposphere (9-12 km altitudes) range from 30 to 200 ppbv, with the highest concentrations reaching close to 200 ppbv, surpassing those in the $0°E$ – $40°E$ longitudinal band.

In the Asian industrial areas (longitudinal band $90°E$ – $150°E$; Fig 5d), CO concentrations display patterns resembling the overall measurements (Fig 5a). A comparison of measurements in western Europe (Fig 5b) with Asian industrial regions (Fig 5d) indicates more elevated CO concentrations above 3 km altitudes in the Asian troposphere than in western European regions. These comparisons suggest that Asian regions are the primary sources of CO in the troposphere, and the upper troposphere is notably impacted by ground-level CO pollution.

Over the north Pacific flight routes (longitudinal bands $150°E$ – $160°W$; Fig 5e), CO concentrations were measured in the 9.5-12 km altitudes, revealing the presence of elevated CO exceeding 200 ppbv. Some data points indicate the highest CO concentrations reaching 600 ppbv around 11 km altitude. In comparison with the upper troposphere downwind of European industrial regions (Fig 5b), the Asian downwind regions of the Pacific upper troposphere clearly exhibit higher CO levels than the downwind regions of European industrial areas. Since CO measured in the upper troposphere over the $0°E$ – $40°E$ longitudinal band is mostly less than 100 ppbv (Fig 5b), the source regions for the elevated CO (close to 200 ppbv and reaching 600 ppbv in some instances) must originate from longitudinal bands between $40°E$ and $150°E$.

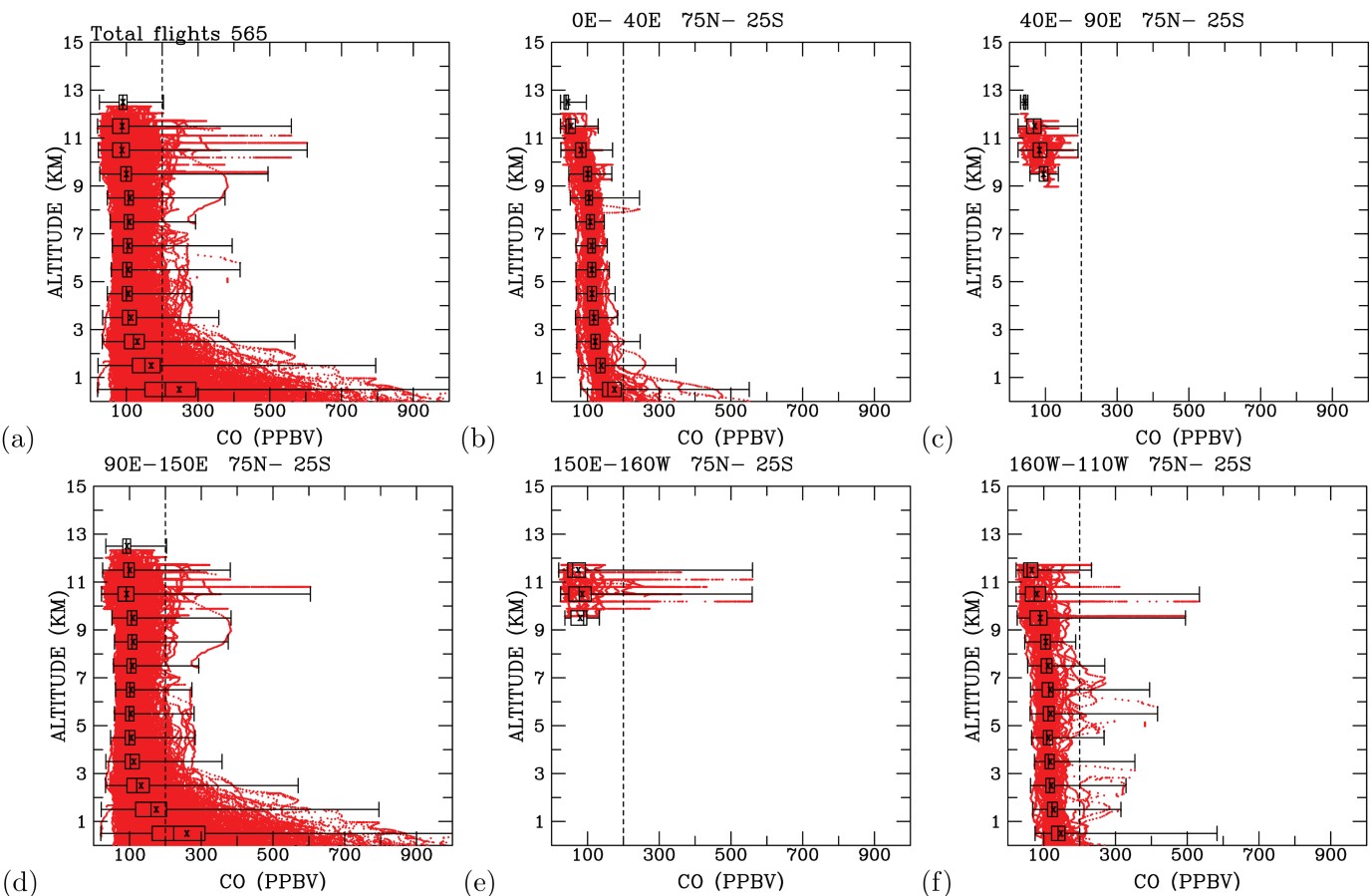

**Fig 5. Profiles of CO measurements.** (a) all flights; and flights in selected longitudinal regions: (b) Western Europe, $0° – 40°E$; (c) downwind of Western Europe, $40° – 90°E$; (d) Asian industrial areas, $90° – 150°E$; (e) downwind of Asian industrial areas and over the Northwest Pacific, $150°E – 160°W$; and (f) the Northeast Pacific, $160°W–110°W$. The black box represents (from left to right) the 25th, 50th (median), and 75th percentiles of all data. The mean of all measurements is marked with an $x$. The horizontal black line indicates the maximum and minimum CO concentrations measured.

For the longitudinal band $160°W – 110°W$ (Fig 5f), profiles of CO measurements closely resemble those measured over western Europe (Fig 5a), except that CO concentrations close to the ground are mostly less than 200 ppbv. However, in the upper troposphere, some measurements indicate concentrations exceeding 200 ppbv, reaching as high as 500 ppbv. The subsequent section delves into determining the sources of these high CO concentrations.

**Time-series CO distribution.** To further explore the origins of elevated carbon monoxide (CO) levels in the upper troposphere, we analyze time-series data for CO measurements within the altitude range of 9 to 12 km. Fig 6 isolates Northern Hemisphere midlatitude data (e.g., $20°N – 70°N$) to highlight North Pacific trends, with altitude traces retained to reflect data density per flight. The time-series over Europe (Fig 6a) depicts CO concentrations varying between 40 and 160 ppbv, with the highest concentrations (close to 160 ppbv) predominantly occurring during the summer months of July to September. Downwind of western Europe (Fig 6b), CO concentrations in the upper troposphere also fluctuate within the range of 40 ppbv to 160 ppbv, resembling the measurements over western Europe. Notably, on two

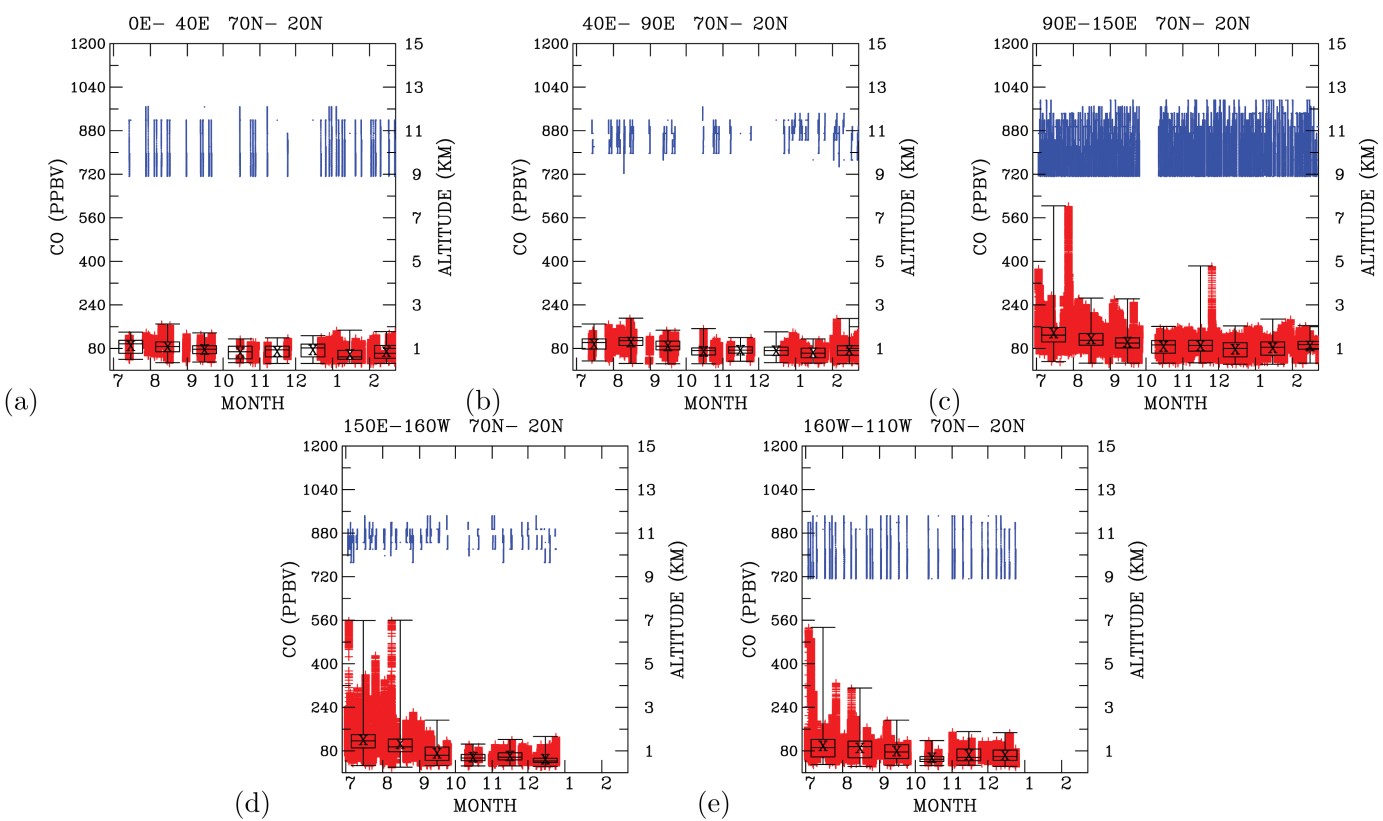

**Fig 6. Time-series CO concentrations above 9-km altitudes in longitudinal regions.** (a) $0° − 40°E$, (b) $40° − 90°E$, (c) $90° − 150°E$, (d) $150°E − 160°W$, and (e) $160°W − 110°W$. Aircraft altitudes are colored in blue (scales show in right y-axis), and CO concentrations are colored in red (scales shown in left y-axis). The black box indicates (from bottom to the top) the $25^{th}$, $50^{th}$, and $75^{th}$ percentiles of all monthly data. The mean of all monthly measurements are marked by a x sign. The vertical black line indicates the monthly maximum and minimum CO concentrations measured.

occasions, CO levels reached 180 ppbv in August 2012 and February 2013. Elevated CO concentrations (close to 160 ppbv) were mainly observed during the summer months of July to August and in September, aligning with the patterns observed in the upper troposphere over western Europe.

Examining the time-series in the upper troposphere over the Asian industrial regions (Fig 6c), CO concentrations ranged between 160 ppbv and 240 ppbv from July to September. From October 2012 to February 2013, CO concentrations remained below 160 ppbv. Three notable instances occurred when CO concentrations reached 360 ppbv (early July), 600 ppbv (late July), and 360 ppbv (late November) in the upper troposphere. Fig 6c clearly illustrates that the upper troposphere of the Asian industrial regions contained 50% more CO than over western Europe from July to September 2012. The upper troposphere during the summer months (July to August) and September contained 50% more CO than during October to November and the winter months (December, January, and February). CO concentrations in the summer-time upper troposphere exceeded those observed in the winter-time upper troposphere, indicating effective vertical transport of CO from the surface to the upper troposphere in summer compared to winter. The PGGM/IAGOS measurements presented here provide insights into the effective vertical transport processes that control pollution distribution in the upper troposphere.

Further examinations of time-series data in the upper troposphere and downwind of the Asian industrial regions, over the central north Pacific (Fig 6d), and northwestern North America (Fig 6e), consistently reveal similar high CO concentrations during summer compared to winter measurements. The CO measurements over the central north Pacific show more frequent occurrences of CO levels exceeding 240 ppbv during July and August than those observed over the Asian industrial regions (Fig 6c). Time-series measurements of CO over northwestern North America show three instances of elevated CO concentrations reaching 520 ppbv (early July) and 320 ppbv (late July and early August).

The time-series analysis demonstrates that CO concentrations in the upper troposphere from mid-autumn (October) to winter (February) are mostly similar, ranging between 40 ppb and 160 ppbv. However, during the summer months (July to August) and in October, the Asian industrial regions and downwind areas are clearly impacted by vertical transport of pollutants from the surface.

The upper troposphere of the central north Pacific contains the most polluted CO for all longitudes east of $0°E$ and west of $110°W$ during the summer (July to August) and in September. Higher CO concentrations over the central north Pacific than over the Asian industrial regions indicate other sources of CO that contributed to elevated CO in the central north Pacific upper troposphere during the summer.

**Comparisons with the NASA aircraft measurements.** Previous airborne measurements of carbon monoxide (CO) were conducted during the PEM-West A mission in September–October 1991 (Figs 7a–b), PEM-West B in February–March 1994 (Figs 7e–f), and TRACE-P in February–April 2001 using a DC-8 aircraft (Figs 7i–j) and a P-3 aircraft (Figs 7k–l). These pioneering missions provided unique CO measurements over the northwestern Pacific during periods of rapid change: in 1991 and 1994, when industrial development in East Asia was just beginning to accelerate, and in 2001, when anthropogenic emissions had risen significantly. The PGGM/IAGOS measurements conducted routinely over a similar region in September–October 2012 (Figs 7c–d), more than a decade after PEM-West A, enable an evaluation of how CO levels over the northwestern Pacific evolved after intensive fossil fuel combustion emissions during the early 21st century.

The entire troposphere during September–October 2012 exhibited higher pollution levels than in 1991. In 1991, most CO concentrations ranged from 50 ppbv to 150 ppbv (Fig 7b), whereas in 2012, concentrations ranged from 50 ppbv to 200 ppbv (Fig 7d). The upper troposphere in 2012 showed increased pollution, with CO concentrations exceeding 200 ppbv, compared to concentrations below 150 ppbv in 1991. In the lower troposphere (below 3 km), maximum CO concentrations increased from approximately 700 ppbv in 1991 to 780 ppbv in 2012. These trends highlight a marked increase in CO levels throughout the troposphere, especially in the upper troposphere.

During February–March 1994 (Fig 7f), elevated CO concentrations (above 200 ppbv) were observed from the surface to 9-km altitudes. The polluted lower troposphere observed in February–March 1994 by PEM-West B (Fig 7f) is consistent with February–April 2001 measurements from the TRACE-P DC-8 (Fig 7j) and P-3 (Fig 7l) missions. These observations confirm the persistence of high CO concentrations in the lower troposphere over the northwestern Pacific during the late winter and early spring periods. PGGM/IAGOS measurements in February 2013 did not cover the Pacific regions, limiting direct comparisons for this period. Future analyses will incorporate flight data collected after 2012 to further explore changes in CO concentrations over the northwestern Pacific.

Comparisons of aircraft measurements across different time periods clearly demonstrate the growing influence of anthropogenic emissions from the Asian continent on the Pacific

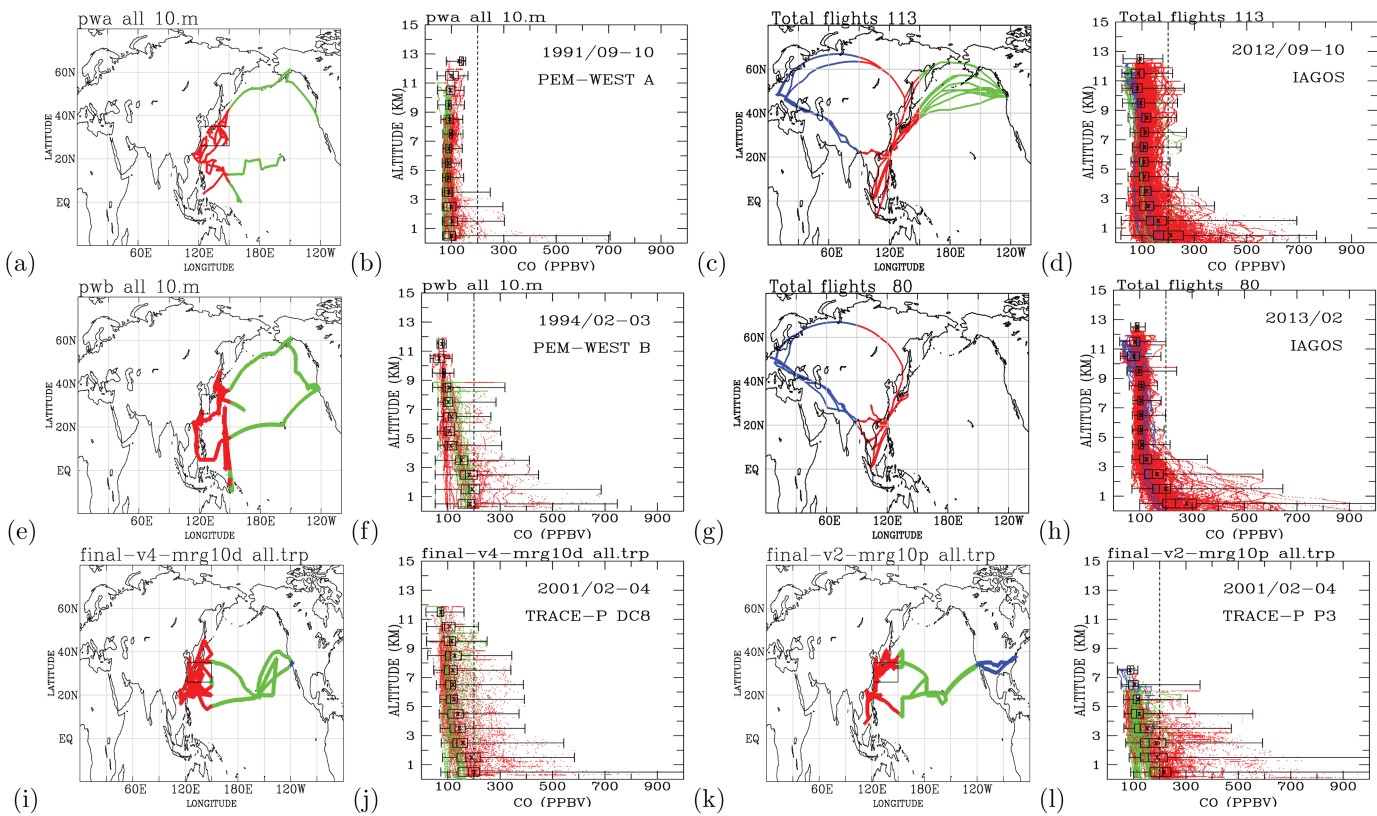

**Fig 7. PEM-West, TRACE-P and IAGOS measurements.** Flight routes (a) and profiles (b) of all CO measurements from PEM-WEST A in September–October 1991; PGGM/IAGOS in September–October 2012, (c) and (d); PEM-WEST B in February–March 1994, (e) and (f); PGGM/IAGOS in February 2013, (g) and (h); and TRACE-P in February–April 2001 from DC-8, (i) and (j), and P-3, (k) and (l). Flight routes are color-coded according to longitudes as follows: blue (west of $90°E$ and east of $120°W$), red (between $90°E$ and $150°E$), and green (between $150°E$ and $120°W$). The vertical black dashed line highlights CO concentrations at 200 ppbv. A dashed line at 200 ppbv underscores a shift from 50–150 ppbv in 1991 to 50–200+ ppbv in 2012, indicating enhanced upper tropospheric pollution. The black box represents the $25^{th}$, $50^{th}$ (median), and $75^{th}$ percentiles of all data (from left to right). The mean of all measurements is marked with an *x*. The horizontal black line indicates the maximum and minimum CO concentrations measured.

troposphere. Regions of the Pacific atmosphere immediately downwind of the Asian continent (between $120°E$ and $150°E$) exhibit higher CO concentrations compared to areas further downwind (east of $150°E$).

In September–October 1991, CO concentrations throughout the Pacific troposphere east of $150°E$ were generally below 100 ppbv (Fig 7b). However, after 21 years of significant industrial growth in Asia, PGGM/IAGOS measurements in the same region during the same months in 2012 show CO concentrations exceeding 100 ppbv below 11-km altitudes. These findings highlight the substantial increase in anthropogenic pollution transported from Asia to the remote Pacific atmosphere over two decades.

The troposphere below 11-km altitudes has experienced increased pollution during the September–October period from 1991 (Fig 7b) to 2012 (Fig 7d). This trend is particularly pronounced during the late winter and early spring months, as evidenced by the comparison of measurements from February–March 1994 (Fig 7f) to February–April 2001 (Fig 7j and 7l). Exceptionally high CO concentrations, exceeding 200 ppbv, were observed in the upper troposphere during measurements taken in 1994 (Fig 7f), 2001 (Figs 7j and 7l), and 2012 (Fig 7d).

## Spatial distribution of CO in the upper troposphere

Elevated concentrations of CO (>100 ppbv) are observed over the North Pacific, East Asia, the tropical northwestern Pacific, east of 80°E longitudes over the Euro-Asian continent, and in north India and Southeast Asia (Fig 8b). No elevated CO concentrations were detected in longitudes west of 80°E.

Low CO concentrations (<50 ppbv) are prevalent over the North Pacific, north of the Euro-Asian continent, and in longitudes west of 80°E to the south of the Euro-Asian continent (Fig 8c). Notably, low CO concentrations are absent over East Asia, the tropical northwestern Pacific, and in longitudes east of 80°E to the south of Asia and Southeast Asia.

## Back trajectories for elevated and low CO in the upper troposphere

Utilizing the observed elevated CO concentrations in the upper troposphere, we conducted Lagrangian back trajectory modeling to pinpoint potential source regions for these heightened CO levels. Elevated CO concentrations over East Asia to the North Pacific were predominantly noted from July to October, whereas elevated CO levels over the Euro-Asian continent and tropical regions were more prevalent from November to February (Fig 9a). The back trajectories of elevated CO during July-October traced back to the Asian continent near the surface, covering latitudinal ranges north of 20°N and longitudinal spans between 60°E and 150°E (Fig 9b). Conversely, the elevated CO concentrations occurring from November 2012 to February 2013 originated from tropical regions and the longitudinal band between 60°E and 150°E (Fig 9a and Fig 9b).

These back trajectories highlight that elevated CO in the North Pacific upper troposphere predominantly originated during summer months [33,34] with convective activity (July and August) and early autumn (September and October) [35–37]. The elevated CO concentrations observed from November to February are linked to convective processes in the tropical northwestern Pacific [38].

Asian anthropogenic emissions and boreal biomass burning dominate CO levels in the North Pacific upper troposphere [39]. Rapid uplift driven by convection and mid-latitude cyclones injects CO into the upper troposphere, facilitating its long-range transport [40,41]. During spring and early summer, efficient vertical transport and favorable wind patterns—such as the Westerlies—lead to heightened CO concentrations. Satellite observations [40], in-situ aircraft measurements [15–17], and models including HYSPLIT [42], FLEXPART [18], and GEOS-Chem [43,44] are frequently employed to study the long-range transport of air pollutants originating from Asian anthropogenic and biomass burning sources. Whereas Jiang et al. [40] used satellite data to illustrate seasonal CO transport, our IAGOS profiles offer in-situ validation with finer temporal resolution over the North Pacific, thereby refining source attribution through backtrajectory analysis.

In the case of very low CO concentrations in the upper troposphere, back trajectory analysis indicates that these low CO levels are primarily associated with air originating north of 40°N (Fig 9d), particularly in the upper troposphere (Fig 9e and Fig 9f). A comparison between high and low CO concentrations clearly underscores surface regions as the primary source of elevated CO in the upper troposphere. The insights gleaned from the analysis presented in Fig 9 suggest that monitoring data from aircraft can effectively unveil sources of air pollution.

To gain further insights into the emission sources and transport pathways hot spots, a gridded analysis of all back trajectories can provide valuable information. In Fig 10a, a 1-degree by 1-degree longitude-latitude analysis of all back trajectories originating from elevated CO

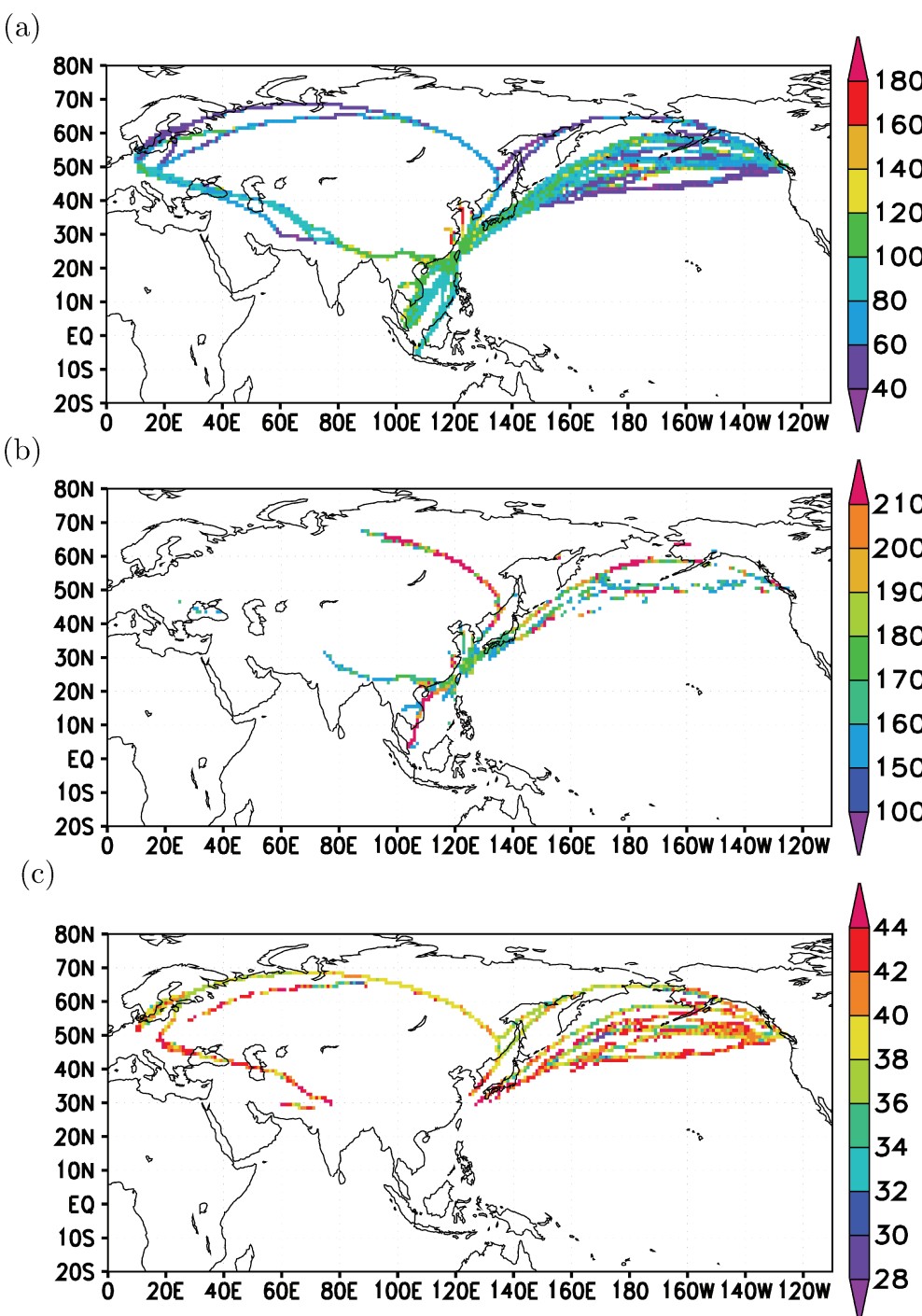

**Fig 8. Mean CO concentrations (in units of ppbv) in altitudes above 9 km.** (a) all measurements; (b) from cases with elevated CO (with the highest limit of 220 ppbv) ; and (c) from cases with low CO (with a lowest limit of 26 ppbv).

observed in the upper troposphere is presented. The trajectories exhibit high densities, particularly concentrating over the eastern part of the Asian continent. Concentrated in latitudes

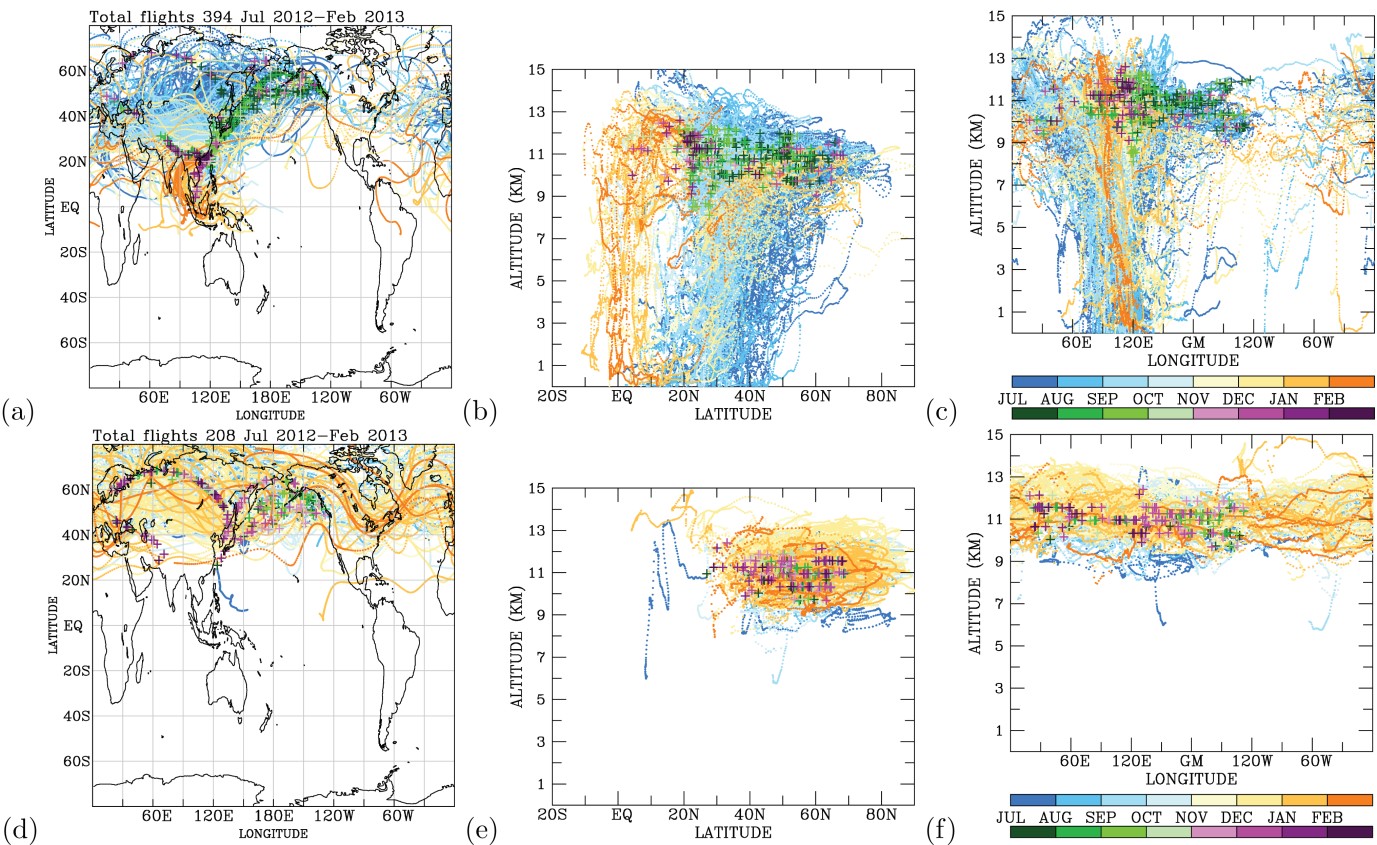

**Fig 9. Back trajectories (lines colored, from blue to orange and according to month) and locations (crosses colored by concentration levels, from green to purple and according to month).** Cases With elevated (a) and low (d) CO in the upper troposphere (above 9-km altitudes). Latitudinal and longitudinal cross-sections of back trajectories for elevated cases of CO are shown in (b) and (c), and in (e) and (f) for low cases of CO.

between the equator and 45°N in the lower troposphere, below 1-km altitude (Fig 10b), these back trajectories vertically extend from 1-km to as high as 13-km altitudes. The vertical axis of the high-density back trajectory occurrences is centered around 30°N, with a narrow distribution between 2-km and 7-km altitudes, widening from 7-km to 11-km and containing the widest latitudinal extent from about 20°N to 60°N. Longitudinally, the vertical extent of high-occurrence trajectory density is centered around 120°E, with ranges between 60°E and 150°E (Fig 10c).

In terms of the spatial distribution of back trajectory occurrences on a 1-degree by 1-degree longitude-latitude grid resolution, for very low CO concentrations in the upper troposphere, the areas are more evenly distributed in high latitudes (Fig 10d), mostly concentrated between 10-km and 13-km altitudes and north of 30 N (Fig 10e). Longitudinally, the distribution densities are more evenly spread compared to the occurrence for elevated CO (Fig 10f).

Fig 10 leverages monitoring data to anticipate hot spot areas of CO emissions. To assess the realism of these predictions, Fig 11a provides a close-up of the hot spot areas, revealing high densities of trajectory passage per 1-degree resolution over East and South Asia. Concentrated over China (in the longitudes between 100°E and 130°E, and in the latitudes between 20°N and 40°N) and northeastern India, these hot spot areas exhibit patterns similar to the

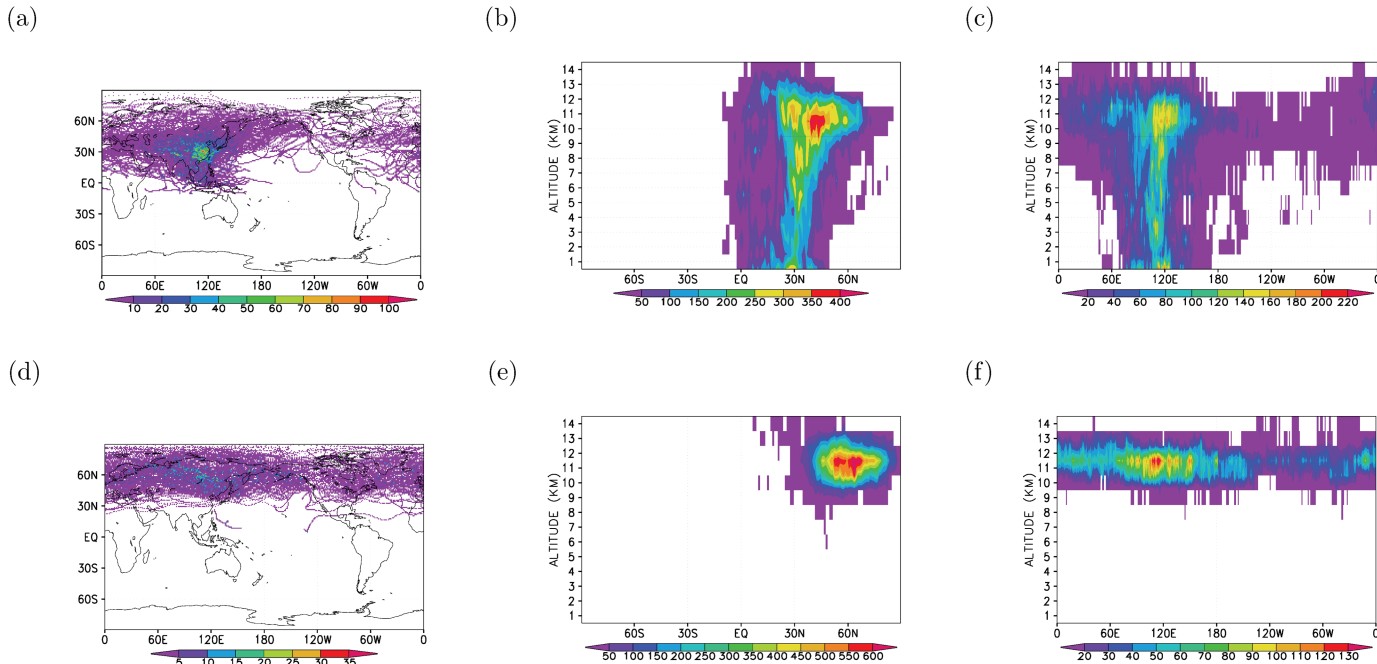

**Fig 10. Back trajectories densities (in units of trajectory points per 1-degree longitude-latitude, or 1-degree altitude cross-sections).** For cases with elevated CO in (a), (b) and (c); for cases with low CO, in (d), (e) and (f).

CO emissions inventory estimated by ACE-Asia P (Fig 11b). Notably, the back trajectory density distribution closely reproduces the circular-shaped CO emissions distribution over mid-Asia (in the longitudes between $70°E$ and $100°E$, and latitudes between $20°N$ and $50°N$) from ACE-Asia P.

Fig 11a underscores that central East Asia (in longitudes between $100°E$ and $130°E$, and in latitudes between $20°N$ and $40°N$) and the regions over northeastern India are particularly effective in exporting surface pollutants to the North Pacific upper troposphere during the study period—a key finding of this work. Averaged CO concentrations per 1-degree resolution, based on observed CO in flights from all back trajectories, reveal that concentrations exceeding 250 ppbv are distributed north of $40°N$ and over tropical regions (Fig 12a). These analyses suggest that high latitudes (north of $40°N$) and the tropical regions are also source areas for elevated CO in the upper troposphere.

When examining back trajectories reaching the lower troposphere (within 1-km altitude from the surface), their concentration is predominantly over East Asia (Fig 12b). This analysis indicates that elevated CO, distributed north of $40°N$ (as depicted in Fig 12a), is mostly confined to the lower troposphere.

The regions in East Asia (between $100°E$ and $130°E$, and between $15°N$ and $40°N$) and over northeastern India emerge as hot spot areas of surface CO contributing to elevated CO in the upper troposphere (Fig 12c), with the monitoring data inferring elevated low-level CO concentrations (Fig 12d).

## Time-series profiles of CO measurements over East Asia

We have demonstrated that the mid-latitudes over East Asia serve as source regions for elevated CO observed in the North Pacific upper troposphere during the study period. With

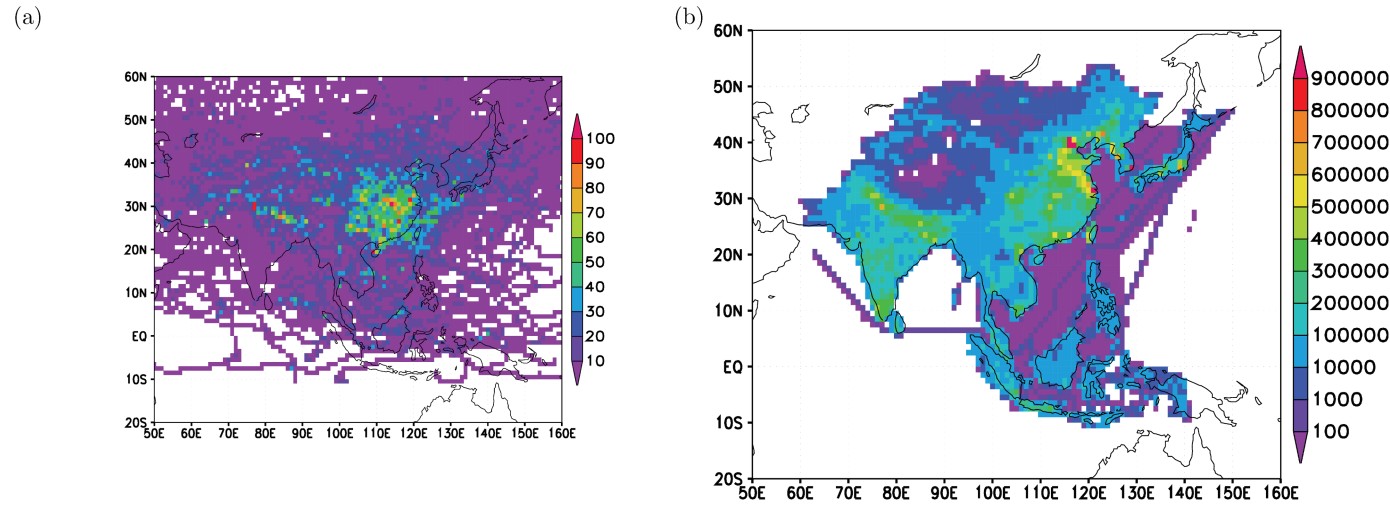

**Fig 11. Back trajectory densities over East Asia (a), and (b) surface CO emissions (in units of ton CO per year per 1-degree latitude-longitude grid) from TRACE-P.**

a total of 521 descending and ascending flights over Taipei, these flights offer the high-frequency profile measurements of CO over a subtropical region (approximately 25N) in East Asia.

Fig 13 illustrates the time-series of all 520 profiles of CO measurements conducted between July 2, 2012, and February 15, 2013, over Taipei. Elevated CO concentrations are frequently observed in the lower troposphere, with the highest concentrations surpassing 175 ppbv in close proximity to the surface. This elevated CO presence extends from the surface to the middle and upper troposphere. During the summer months of July and August (Fig 13a), high CO concentrations (>175 ppbv) are predominantly confined to below 2-km altitudes. The vertical extent of high CO expands to 2-3 km altitudes during September-October (Fig 13b) and further extends to 3-4 km altitudes (Fig 13c).

These profiles indicate that the lower troposphere (below 3-km altitude) is largely immersed in elevated CO after August 30, 2012, signifying a highly polluted lower troposphere that can persist for several months. Prior to August 30, the lower troposphere below 2-km altitudes may occasionally exhibit cleaner air (with CO concentrations less than 125 ppbv). Moreover, these profiles, utilizing CO as a reliable tracer, reveal the vertical extent of a polluted troposphere evolving from summer to winter.

## Back trajectories for elevated CO in the lower troposphere

To discern the origins of polluted and clean air in the lower troposphere, we performed model calculations to pinpoint their sources. The model simulations reveal that the polluted air, characterized by elevated CO concentrations at 2-km altitude, predominantly originated from latitudes north of 20N and longitudes west of $120°E$, with a primary occurrence during the autumn to winter months (Figs 14a-c). In contrast, trajectories of cleaner air at 2-km altitude indicate that this pristine air predominantly originated from tropical regions (south of $20°N$) and east of $120°E$ (Figs 14d-f). Fig 13 presents high-time resolution lower tropospheric profiles over Taipei, while Fig 14 reveals distinct source regions for elevated versus low CO, complementing the upper tropospheric focus of Figs 5–7.

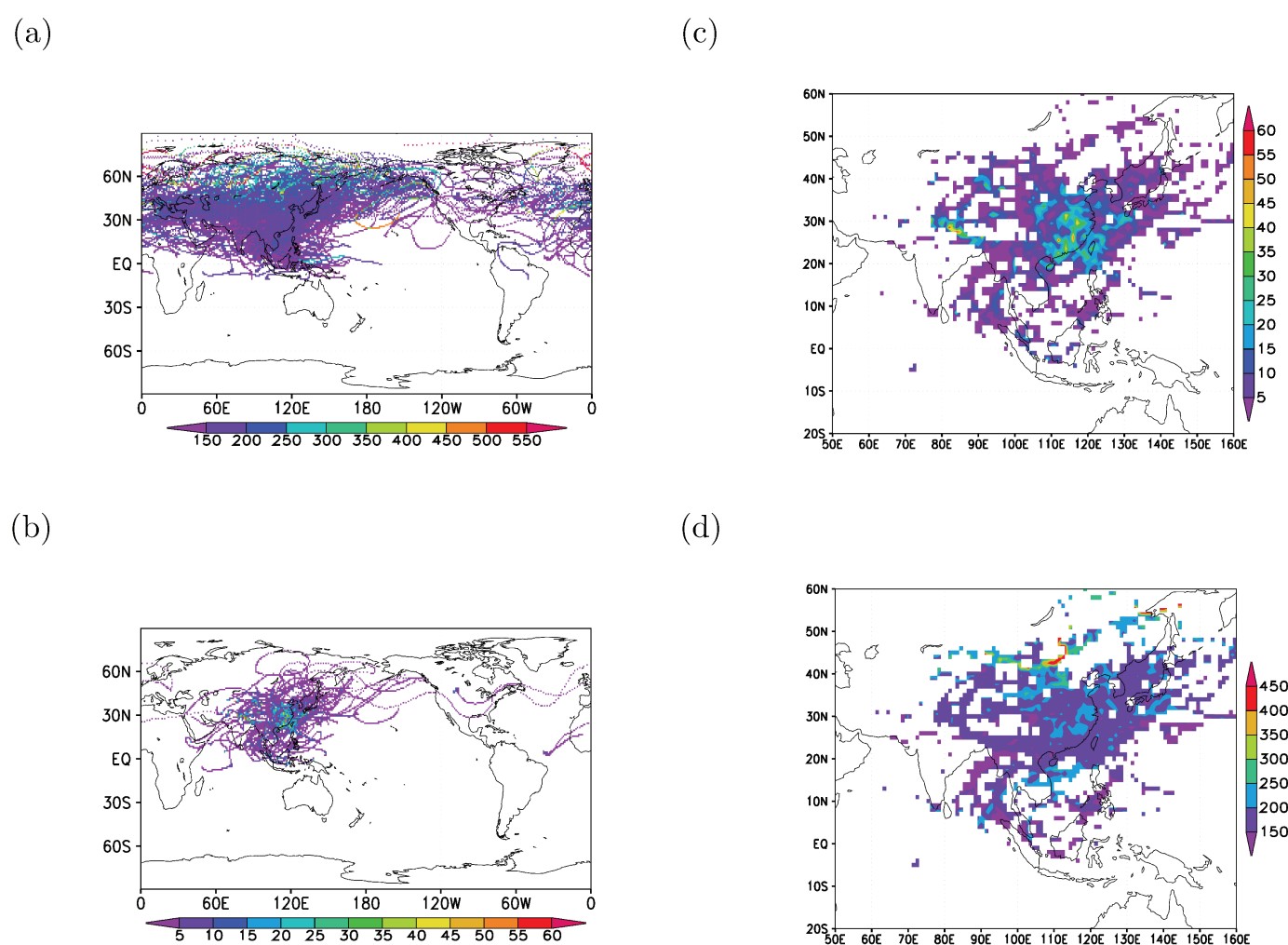

**Fig 12. Gridded analysis of back trajectories associated with elevated CO concentrations.** (a) Mean CO concentrations determined from elevated CO measurements and back trajectories (units: ppbv); (b) density of back trajectories reaching below 1-km altitude (frequencies per 1-km square grid box); (c) density of back trajectories reaching below 1-km altitude over East Asia (frequencies per 1-km square grid box); (d) mean CO concentrations from back trajectories reaching below 1-km altitude (units: ppbv).

## Summary

The upper troposphere over the North Pacific has experienced a significant increase in CO pollution originating from surface sources over the 21-year period from 1991 to 2013. This upward trend is substantiated by PGGM/IAGOS aircraft measurements conducted between June 2012 and February 2013 and compared with NASA aircraft missions in 1991, 1994, and 2001 [32]. The analysis reveals a consistent rise in anthropogenic pollution in the North Pacific upper troposphere over the two decades, particularly downwind of China. These findings align with the observed increase in nitrogen oxide concentrations over China reported from 1996 to 2004.

Key observations indicate that the troposphere downwind of the Asian continent has become increasingly polluted, as has the North Pacific upper troposphere [45]. Conversely,

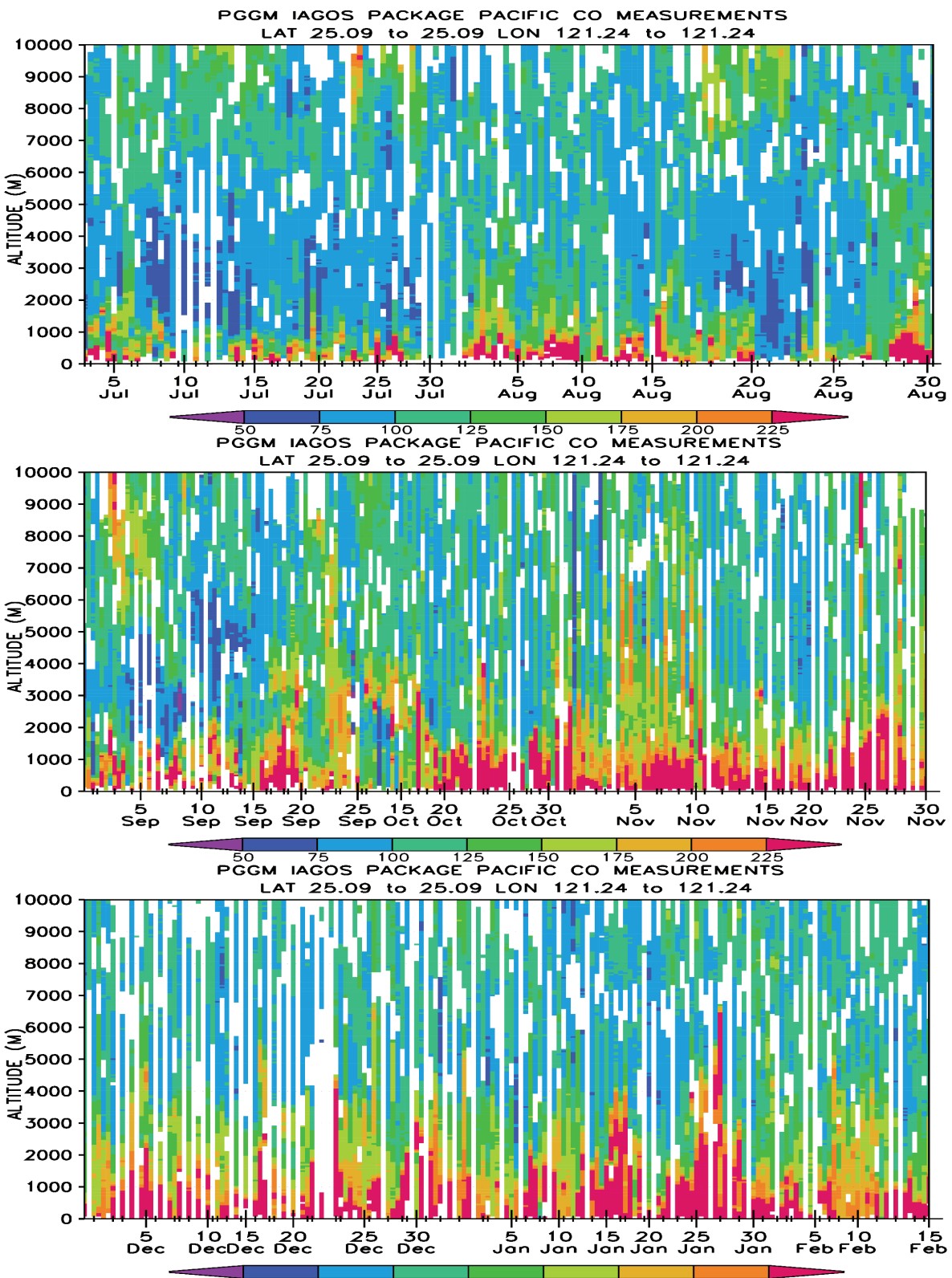

**Fig 13. Time-series profiles of CO measurements (in units of ppbv and at 50-m vertical resolution) for flights ascending and descending to Taipei from 2 July 2012 to 15 February 2013.**

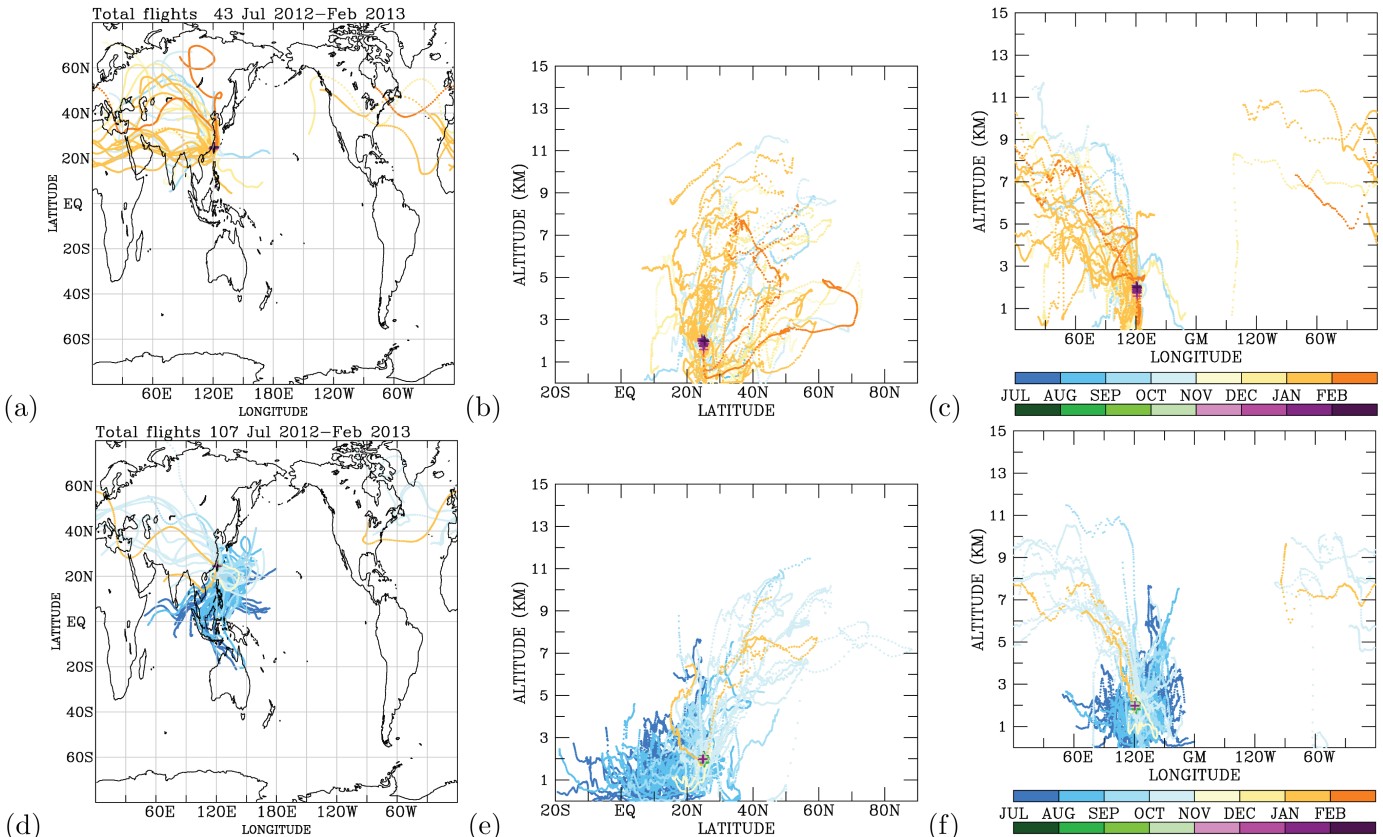

**Fig 14. Back trajectories for cases with elevated CO (greater than 100 ppbv; top panels) and low CO (less than 45 ppbv; bottom panels) at 2-km altitude from Taipei.** (a) and (d) show the longitude-latitude distribution of back trajectories. (b) and (e) display the latitude-altitude distribution. (c) and (f) present the longitude-altitude distribution. The top color bar indicates the time in months for the back trajectories, while the bottom color bar shows the time in months for elevated CO (c) and low CO (f) when the back trajectories were initiated.

upwind regions of the Asian continent exhibit lower pollution levels compared to their downwind counterparts [46]. Exceptionally high CO concentrations observed in the upper troposphere, as demonstrated in this study, suggest effective vertical transport of ground-level anthropogenic pollutants—a phenomenon less pronounced over Europe. Fig 7 presents a statistical comparison of CO measurements from the NASA GTE experiments and IAGOS data. The PEM-West A measurements from 1991 indicate mean CO concentrations of approximately 100 ppbv, with occasional elevated concentrations exceeding 200 ppbv at altitudes below 4 km. In contrast, the IAGOS measurements from 2012 show mean CO concentrations exceeding 100 ppbv, with elevated CO frequently surpassing 200 ppbv throughout the troposphere. These findings highlight the temporal evolution of CO concentrations over two decades and emphasize the differences between the historical NASA GTE experiments and the more recent IAGOS observations. Long-term trends of CO in the upper troposphere upstream of Asian industrial areas have been examined in a separate study [32].

Distinct seasonal variations characterize the downwind regions of the Asian continent and the North Pacific upper troposphere. Elevated CO concentrations, exceeding 160 ppbv, are prevalent during the summer months of July and August, with a subsequent decline in September. Conversely, CO concentrations remain low, below 100 ppbv, from October 2012

to February 2013. These seasonal patterns are less conspicuous in the upwind areas of the Asian industrial regions.

Consistent with previous observations over northeast Asia in 2003 and North America, elevated CO concentrations in the upper troposphere during summer are associated with convective transport. HYSPLIT model back trajectory calculations further confirm that ground-level Asian industrial regions are the primary sources of CO pollution in the downwind regions of the North Pacific upper troposphere. Conversely, the back trajectory calculations for low CO concentrations in the North Pacific upper troposphere do not trace back to the ground-level Asian industrial regions.

## Acknowledgments

We extend our sincere gratitude to China Airlines (CAL) and Evergreen Marine Corporation (EMC) for their participation in the PGGM/IAGOS project. We also gratefully acknowledge financial support from the Taiwan Ministry of Science and Technology (now the National Science and Technology Council) under grant 96WFA0700420 and the Environmental Protection Administration (now the Ministry of Environment) for funding the PGGM project. Additionally, we thank the European Council for its support in funding the IAGOS project. The funders had no role in the study design, data collection and analysis, decision to publish, or preparation of the manuscript. We acknowledge the use of the NOAA HYSPLIT model and the PEM-WEST A, PEM-WEST B, and TRACE-P datasets provided by NASA.

## Author contributions

**Conceptualization:** Kuo-Ying Wang.

**Data curation:** Kuo-Ying Wang, Philippe Nedelec.

**Formal analysis:** Kuo-Ying Wang.

**Funding acquisition:** Kuo-Ying Wang, Valerie Thouret, Hannah Clark, Andreas Wahner, Andreas Petzold.

**Investigation:** Kuo-Ying Wang, Hannah Clark, Andreas Wahner, Andreas Petzold.

**Methodology:** Kuo-Ying Wang, Philippe Nedelec, Andreas Wahner, Andreas Petzold.

**Project administration:** Kuo-Ying Wang, Valerie Thouret, Hannah Clark, Andreas Wahner, Andreas Petzold.

**Resources:** Kuo-Ying Wang, Philippe Nedelec, Valerie Thouret, Hannah Clark, Andreas Wahner, Andreas Petzold.

**Software:** Kuo-Ying Wang.

**Supervision:** Kuo-Ying Wang, Philippe Nedelec, Valerie Thouret, Hannah Clark, Andreas Wahner, Andreas Petzold.

**Validation:** Kuo-Ying Wang, Philippe Nedelec, Valerie Thouret, Hannah Clark, Andreas Wahner, Andreas Petzold.

**Visualization:** Kuo-Ying Wang.

**Writing – original draft:** Kuo-Ying Wang.

**Writing – review & editing:** Kuo-Ying Wang, Valerie Thouret, Hannah Clark, Andreas Wahner, Andreas Petzold.

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
