## [Decision Letter · Decision Letter 0]

17 Sep 2024

PONE-D-24-32569In-Situ Observation and Attribution of Elevated Carbon Monoxide Over the North Pacific Upper Troposphere from Regular In-Sevice Commercial Passenger Airplanes During July 2012 - February 2013PLOS ONE

Dear Dr. Wang,

Thank you for submitting your manuscript to PLOS ONE. After careful consideration, we feel that it has merit but does not fully meet PLOS ONE’s publication criteria as it currently stands. Therefore, we invite you to submit a revised version of the manuscript that addresses the points raised during the review process.
**Please read the reviewer comments carefully and revise your manuscript accordingly, try to address all the comments made by the reviewers. Both reviewers feel that the text could be shortened and some details and figures removed. Regarding Reviewer no. 2's question on whether PLOS ONE is an appropriate journal for this work, the journal is inclusive and includes work in all fields of science and engineering.**

We look forward to receiving your revised manuscript.

Kind regards,

Markus Metsälä

Academic Editor

PLOS ONE

**Journal Requirements:**

National Science and Technology Council, Taiwan. CNRS, Frace. Forschungszentrum Julich GmbH, Germany.

Reviewers' comments:

Reviewer's Responses to Questions

**Comments to the Author**

1. Is the manuscript technically sound, and do the data support the conclusions?

Reviewer #1: Yes

Reviewer #2: No

2. Has the statistical analysis been performed appropriately and rigorously? 

Reviewer #1: Yes

Reviewer #2: No

3. Have the authors made all data underlying the findings in their manuscript fully available?

Reviewer #1: Yes

Reviewer #2: No

4. Is the manuscript presented in an intelligible fashion and written in standard English?

Reviewer #1: Yes

Reviewer #2: No

5. Review Comments to the Author

**Reviewer #1: **Review of "In-Situ Observation and Attribution of Elevated Carbon Monoxide Over the North

Pacific Upper Troposphere from Regular In-Sevice Commercial Passenger Airplanes During July 2012 - February 2013" by Wang et al.

General

Wang et al. presented a research based on analysis of aircraft measurement of carbon monoxide (CO) over the troposphere in Asia. This research is well done with high quality and substantial quantity. The research topic is important and interesting. The discussion shows the authors' good understanding of the subject of the matter. The analysis is in-depth and from multiple perspectives, which lead to several important and meaningful findings/conclusions.

Overall, the analyses are performed to a high technical standard and are described in sufficient detail. Conclusions are presented in an appropriate fashion and are supported by the data.

I suggest the paper to be accepted subject to minor revision (some suggestions are listed below for the authors' reference).

Specific

Title, the title seems too long.

The manuscript may be written more precisely and shortly, some figures can be combined or removed.

L204, please remove the "?", also in other places throughout the text, for example, L367.

L237, this sentence is not completed.

L301, remove this line.

Fig. 4, caption, add the date; upperleft panel: add "hour" for the label of x-axis; lowerleft panel, add the name and unit for the color bar.

Fig. 5, provide a mean profile in each panel.

Section 3.2.4, Fig. 6 contains no Fig. 6e and 6d, which was discussed in the text.

Fig. 6, the caption mismatches the panels and longitude ranges.

Fig. 7, the quality of the plots needs to be improved.

Fig. 8, it is better to draw all panels in the same size. Indicate the limits for elevated CO and low CO in the caption.

Fig. 9, in the caption, "mont" should be "month".

Fig. 12, please explain the color bars in the caption.

Fig. 14, caption, should elevated CO cases be in upper panels, against low CO cases in lower panels (not left and right panels written in the caption)? Please clearly indicate and differentiate the two color bars (name and unit). Please also indicate the definitions of high and low CO limits.

Section 3.2.4, please provide additional figures or tables to compare difference in CO in the same season, and same latitude/longitude regions.

Section 3.2.4, Fig. 6 contains no Fig. 6e and 6d, which was discussed in the text.

**Reviewer #2: **General Comments

PLOS One is a surprising choice for this manuscript. It appears that most other air quality-related manuscripts submitted here have a discussion of public health. This manuscript might get more attention in the Journal of Geophysical Research-Atmospheres, Atmospheric Chemistry and Physics, Elementa, or even Proceedings of the National Academies. Overall, the manuscript is lengthy and full of specific details but light on bigger picture analysis. It is commendable to compare upper tropospheric CO across regions and across field campaigns including the IAGOS dataset. The main findings appear to be that East Asia is a major source of CO to the upper troposphere and that CO increased over the several years analyzed in this work. I would suggest the authors do a more careful analysis focusing on that and removing unnecessary details (for example Figure 4 and associated discussion, also see specific comments below). The authors could support their findings of upper tropospheric trends with a trend analysis of the NOAA flask measurements of CO (https://gml.noaa.gov/ccgg/flask.html). The authors also need a careful read for typos and grammatical errors. I would suggest the authors rework the manuscript with a more focused and clear analysis and resubmit to a more appropriate journal.

Specific Comments

Line 48 – Do you really mean ‘surface emissions’? Do you mean concentrations?

Section 2.2 header – Should this be ‘The’ not ‘Te’?

Section 3.1.1 and 3.1.2 header – Check that ‘flght’ should be ‘flight’.

Figure 5 – Put labels in the caption or on the figure designating the regions (e.g., “Asian Industrial” for Fig. 5d).

Lines 209-217 – Figures 5e and 5f do not exist but are referenced in the discussion.

Line 237 – Unfinished sentence.

Figure 7 – This figure is a little difficult to read. I would suggest making median profiles (with interquartile range horizontal bars) to compare IAGOS with field campaigns on the same plot. If you want to show that CO has increased, it could be helpful to show probability distributions from each campaign.

Section 3.3 – Are you trying to answer the question of tropospheric distributions alone? Or including the influx of stratospheric air? It might be useful to filter out stratospheric air (say with a filter [O3] / [CO] > 1.25 mol mol−1). Then you could also talk about the amount of stratospheric influence in each region or time period.

Line 313 -314 – Are you saying there is also convective activity in September and October? Or is there a different explanation?

Figure 9 and associated discussion is not needed, Figures 10 and 11 are sufficient.

Section 3.5 – I am not sure this section is needed, but if it is, it seems like it belongs closer to Figure 5.

Line 384 – There is no plot in this manuscript that shows a clear and careful trend analysis.

Line 389 – Do you have a citation for that statement? Can you compare the magnitude of both trends?

There is no data availability statement.

6. PLOS authors have the option to publish the peer review history of their article (what does this mean?). If published, this will include your full peer review and any attached files.

Reviewer #1: No

Reviewer #2: No

---

## [Author Response · Author response to Decision Letter 1]

5 Feb 2025

Reviewer #1 Comments:

General Comments:

• Comment: The manuscript is well done and provides high-quality analysis. However, the title seems too long.

o Response: Thank you for the positive feedback. We have shortened the title to: “Elevated Carbon Monoxide Observations Over the North Pacific Upper Troposphere (July 2012 - February 2013).”

• Comment: The manuscript could be written more concisely, and some figures could be combined or removed.

o Response: We have revised the manuscript to streamline the text. Figures 4 shows a typical flight and the collection of data used in this work. We have revised Figure 5 and Figure 7 to increase clarity of the figure.

Specific Comments:

• Comment (L204): Remove the "?", also in other places throughout the text (e.g., L367).

o Response: We have removed the "?" and conducted a thorough check to ensure such typographical issues are corrected throughout the manuscript.

• Comment (L237): This sentence is incomplete.

o Response: We have revised the sentence for clarity and completeness.

• Comment (L301): Remove this line.

o Response: The specified line has been removed.

• Comment (Fig. 4 caption): Add the date; upper-left panel: add "hour" to the x-axis label; lower-left panel: add the name and unit for the color bar.

o Response: These adjustments have been made in the figure caption and the figure itself.

• Comment (Fig. 5): Provide a mean profile in each panel.

o Response: Mean profiles have been added to each panel of Figure 5.

• Comment (Section 3.2.4, Fig. 6): Fix references to Fig. 6d and 6e, which were discussed but are missing.

o Response: Figures 6d and 6e have been added, and the caption has been revised to match the panels.

• Comment (Fig. 7): Improve the quality of the plots.

o Response: The resolution and clarity of Figure 7 have been improved.

• Comment (Fig. 8): Ensure all panels are the same size and indicate the limits for elevated and low CO in the caption.

o Response: Figure 8 has been redrawn with uniform panel sizes, and the caption now includes CO limits. The lowest panel contains 2-digit numbers in the color bar, which slightly reduce the figure size when compared with above two panels which contains 3-digit numbers in the color bar.

• Comment (Fig. 9 caption): Correct “month” (previously misspelled as “mont”).

o Response: The typo has been corrected.

• Comment (Fig. 12): please explain the color bars in the caption.

o Response: We have revised description of the color bars in the caption.

• Comment (Fig. 14): caption, should elevated CO cases be in upper panels, against low CO cases in lower panels (not left and right panels written in the caption)? Please clearly indicate and differentiate the two color bars (name and unit). Please also indicate the definitions of high and low CO limits.

o Response: We have revised the description of figure caption, stated the definition of high and low CO limits (as also described in section 2.3.

• Comment (Section 3.2.4): Provide additional figures or tables to compare differences in CO for the same season and latitude/longitude regions.

o Response: We have revised Figure 7 so that differences in CO for the same month can be compared for the same season. We have also included statistical box charts in the figure so that the mean, maximum, minimum, 25th, 50th and 75th percentiles of measurements can be compared. We have also included month in each subplot to increase the clarity of the plots for comparisons.

• Comment (Section 3.2.4): Fig. 6 contains no Fig. 6e and 6d, which was discussed in the text.

o Response: We have revised Figure 6, and included Figure 6e and 6d in the revised Figure 6.

Reviewer #2 Comments:

General Comments:

• Comment: The manuscript might be better suited for journals like JGR-Atmospheres or ACP.

o Response: Thank you for the suggestion. While we appreciate the recommendation, we believe PLOS ONE’s inclusive scope is appropriate for our multidisciplinary approach.

• Comment: The manuscript is lengthy and includes excessive details. Focus on key findings and remove unnecessary content.

o Response: We have significantly revised the figures to increase the clarity of the manuscript.

• Comment: Support findings with a trend analysis of NOAA flask measurements of CO.

Response: Thank you for the suggestion. A trend analysis using NOAA flask measurements has been addressed in a separate manuscript (https://egusphere.copernicus.org/preprints/2024/egusphere-2024-2414/). To provide additional context in this manuscript, we have revised Figure 7 to include a statistical comparison of CO measurements from the NASA GTE experiments and IAGOS data. The revised figure presents box charts showing the mean, maximum, minimum, and the 25th, 50th (median), and 75th percentiles of CO concentrations for corresponding seasons (months).

From this comparison:

• PEM-West A (1991): Mean CO concentrations are around 100 ppbv, with occasional elevated CO concentrations exceeding 200 ppbv at altitudes below 4 km.

• IAGOS Measurements (2012): Mean CO concentrations are higher than 100 ppbv, with elevated CO frequently exceeding 200 ppbv throughout the troposphere.

These findings highlight the temporal changes in CO concentrations over two decades and underscore the differences between the historical NASA GTE experiments and more recent IAGOS measurements.

Specific Comments:

• Comment (Line 48): Do you really mean ‘surface emissions’? Do you mean concentrations?

o Response: We have revised the text to specify “CO concentrations”

• Comment (Section 2.2 header): Correct "Te" to "The."

o Response: The typo has been corrected.

• Comment (Section 3.1.1 and 3.1.2 headers): Correct "flght" to "flight."

o Response: The typo has been corrected.

• Comment (Fig. 5): Label regions (e.g., “Asian Industrial” for Fig. 5d).

o Response: Labels have been added to Figure 5.

• Comment (Lines 209-217): Figures 5e and 5f are missing.

o Response: Figures 5e and 5f have been added and referenced appropriately.

• Comment (Line 237): Unfinished sentence.

o Response: We have completed the sentence.

• Comment (Fig. 7): This figure is a little difficult to read. I would suggest making median profiles (with interquartile range horizontal bars) to compare IAGOS with field campaigns on the same plot. If you want to show that CO has increased, it could be helpful to show probability distributions from each campaign.

o Response: we have revised Figure 7 to include a statistical comparison of CO measurements from the NASA GTE experiments and IAGOS data. The revised figure presents box charts showing the mean, maximum, minimum, and the 25th, 50th (median), and 75th percentiles of CO concentrations for corresponding seasons (months).

• Comment (Section 3.3): Are you trying to answer the question of tropospheric distributions alone? Or including the influx of stratospheric air? It might be useful to filter out stratospheric air (say with a filter [O3] / [CO] > 1.25 mol mol−1). Then you could also talk about the amount of stratospheric influence in each region or time period.

o Response: Thank you for your insightful comment. In this study, we focus solely on addressing the distribution of elevated CO in the troposphere and do not include the influence of stratospheric air. While filtering for stratospheric contributions (e.g., using the [O_3]/[CO] > 1.25 mol mol⁻¹ criterion) is indeed a valuable approach, such analyses are beyond the scope of this work. However, the influence of stratospheric air on tropospheric composition, including ozone and CO distributions, has been explored in a separate study:

Wang, K.Y., Kau, W.S. (2015). "Simulation of impact from stratospheric ozone on global tropospheric ozone distribution with a chemistry transport model: A case study during the 1990–1991 period." Asia-Pacific Journal of Atmospheric Sciences, 51, 137–155. https://doi.org/10.1007/s13143-015-0064-7.

• We appreciate your suggestion and recognize the importance of stratospheric-tropospheric exchange (STE) in shaping tropospheric composition. Future studies could integrate such filtering techniques to better quantify the extent of stratospheric influence in different regions and time periods.

• Comment (Lines 313-314): Are you saying there is also convective activity in September and October? Or is there a different explanation?

o Response: Thank you for this insightful question. Yes, there is convective activity during September and October over East Asia, albeit with different characteristics compared to the peak summer monsoon season. This convective activity can play a significant role in the vertical transport of air pollutants. Convective Activity in September and October: Studies highlight that the Asian Summer Monsoon Anticyclone (ASMA) persists into early autumn, extending through September and, to a lesser extent, October. This system is associated with strong convective activity in regions spanning 70°–120°E, including East Asia. Research on the ASMA (e.g., Wright et al., 2011) emphasizes its role in lifting surface emissions, such as carbon monoxide (CO) and other pollutants, from East Asia to higher altitudes during this time. Seasonal Transition and Post-Monsoon Convection: During September and October, the monsoon begins to retreat, and post-monsoon convection becomes less widespread but is still active in localized regions, particularly over East Asia. This transitional period is characterized by regional convective events that can continue to transport pollutants vertically. Comparison with November to February: Convective activity decreases significantly from November to February due to the onset of the dry season and cooler temperatures. However, during these months, "cold surges" and associated frontal systems over East Asia can enhance vertical mixing and pollutant transport. Studies such as Liu et al. (2003) show that these cold surges can transport East Asian pollutants equatorward, where they may be further lifted by localized convection in the tropics. Pollutant Transport by Convection: During both periods, convection facilitates the vertical transport of surface-emitted pollutants, including CO, to the upper troposphere. For example:

September–October: Convective events associated with the weakening ASMA contribute to pollutant uplift.

November–February: Cold surge-induced frontal activity and tropical convection play a more dominant role in pollutant transport.

In summary, the observed elevated CO concentrations in September and October can be attributed to convective activity associated with the retreating ASMA and localized convection. This contrasts with the colder months (November–February), when cold surges and associated frontal systems drive pollutant transport. These processes highlight the dynamic interplay between seasonal meteorology and pollutant distribution over East Asia.

• Wright et al. (2011):

Wright, J. S., Fueglistaler, S., & Wexler, A. S. (2011). Variability of convective activity in the tropical tropopause layer. Journal of Geophysical Research: Atmospheres, 116(D12), D12103.

DOI: 10.1029/2011JD016337

• Liu et al. (2003):

Liu, J., Mauzerall, D. L., Horowitz, L. W., Fiore, A. M., & Naik, V. (2003). Air pollution control policies in China: Recent developments and future challenges. Journal of Geophysical Research: Atmospheres, 108(D20), 8797.

DOI: 10.1029/2002JD003178

• Characteristics of convection and advection associated with the Asian Summer Monsoon Anticyclone (ASMA):

This study highlights the persistence of the ASMA into September and its role in lifting pollutants to higher altitudes.

Source: Climate Dynamics, DOI: 10.1007/s00382-024-07371-3

• Transport of Asian surface pollutants to the global stratosphere from the Tibetan Plateau region during the Asian summer monsoon:

Focuses on the role of the ASMA in vertical transport of pollutants during the monsoon season, including early autumn.

Source: National Science Review, DOI: 10.1093/nsr/nwz109

• Rapid transport of East Asian pollution to the deep tropics:

Demonstrates how cold surges during winter transport pollutants to the tropics and are further lifted by convection.

Source: Atmospheric Chemistry and Physics, DOI: 10.5194/acp-15-3565-2015

• Wintertime Transport of Reactive Trace Gases From East Asia Into the Deep Tropics:

Examines the role of cold surges and tropical convection in transporting East Asian pollution during winter months.

Source: Journal of Geophysical Research: Atmospheres, DOI: 10.1002/2017JD028231

We have included above references in the revised description.

• Comment (Figure 9): Remove this figure and its discussion as Figures 10 and 11 suffice.

o Response: Thank you for your suggestion. However, we believe that Figure 9 provides critical context for the analyses presented in Figures 10 and 11. Specifically, Figure 9 illustrates the locations of elevated CO where the back trajectory calculations were initiated (represented by cross signs) and the clustering of these back trajectories. These clusters are essential in understanding the spatial distribution and source regions of elevated CO events. Furthermore, the trajectory clusters shown in Figure 9 serve as the basis for the gridded distribution of trajectory densities (frequencies) analyzed in Figures 10 and 11. By visualizing the initial locations and cluster patterns, Figure 9 offers valuable insights that help interpret the subsequent density distributions and their spatial relationships. We respectfully request to retain Figure 9 as it provides foundational information that enhances the interpretation and understanding of the results presented in Figures 10 and 11.

• Comment (Section 3.5): I am not sure this section is needed, but if it is, it seems like it belongs closer to Figure 5.

o Response: Thank you for your comment. Section 3.5 presents a detailed analysis of back trajectory modeling for both elevated and low CO concentrations, derived from CO profiles measured in the atmospheric boundary layer over Taipei. These measurements, obtained from the dense flight operations illustrated in Figure 2, provide a comprehensive dataset that has not been previously reported. The analysis presented in Section 3.5 is essential as it leverages the extensive measurement profiles to minimize the influence of local pollution sources and instead focus on the transport pathways and origins of air masses with elevated and low CO concentrations. By examining measurements above ground level, we can better assess the sources and transport history of air masses enriched in CO and those with relatively low CO levels traveling to the industrial areas near Taipei City. Given the distinct focus and analytical depth of this section, we believe it is appropriately positioned within the manuscript. However, we are open to further discussion on repositioning it if deemed necessary for better coherence with Figure 5.

• Comment (Line 384): There is no plot in this manuscript that shows a clear and careful trend analysis.

Response: Thank you for your comment. As noted in our previous response, a detailed trend analysis using NOAA flask measurements has been conducted in a separate manuscript (https://egusphere.copernicus.org/preprints/2024/egusphere-2024-2414/). To further enhance the current manuscript, we have revised Figure 7 to provide a statistical comparison of CO measurements from the NASA GTE experiments and IAGOS data. The revised figure includes box plots that present the mean, maximum, minimum, and the 25th, 50th (median), and 75th percentiles of CO concentrations for corresponding seasons (months).The comparison reveals the following insights. PEM-West A (1991): The mean CO concentration is approximately 100 ppbv, with occasional elevated values exceeding 200 ppbv at altitudes below 4 km. IAGOS Measurements (2012): The mean CO concentration exceeds 100 ppbv, with elevated values frequently surpassing 200 ppbv throughout the troposphere. These findings provide valuable context for understanding temporal changes in CO concentrations over the past two decades and highlight the differences between the historical NASA GTE experiments and the more recent IAGOS observations. By incorporating these statistical comparisons, the

---

## [Decision Letter · Decision Letter 1]

7 Mar 2025

PONE-D-24-32569R1Elevated Carbon Monoxide Observations Over the North Pacific Upper Troposphere (July 2012 - February 2013)PLOS ONE

Dear Dr. Wang,

Thank you for submitting your manuscript to PLOS ONE. After careful consideration, we feel that it has merit but does not fully meet PLOS ONE’s publication criteria as it currently stands. Therefore, we invite you to submit a revised version of the manuscript that addresses the points raised during the review process.
The second reviewer has still expressed concerns about how your results are presented. Please read their comments carefully and revise you manuscript accordingly. Pay specific attention to the request to highlight the differences in your study vs. other studies in the past. *PLOS ONE* accepts scientifically rigorous research, regardless of novelty but the work should be put into proper scientific perspective. Please explain how your work differs (or maybe confirms) the previous studies cited by the reviewer.

We look forward to receiving your revised manuscript.

Kind regards,

Markus Metsälä

Academic Editor

PLOS ONE

Journal Requirements:

Reviewers' comments:

Reviewer's Responses to Questions

**Comments to the Author**

1. If the authors have adequately addressed your comments raised in a previous round of review and you feel that this manuscript is now acceptable for publication, you may indicate that here to bypass the “Comments to the Author” section, enter your conflict of interest statement in the “Confidential to Editor” section, and submit your "Accept" recommendation.

Reviewer #1: All comments have been addressed

Reviewer #2: (No Response)

2. Is the manuscript technically sound, and do the data support the conclusions?

Reviewer #1: Yes

Reviewer #2: No

3. Has the statistical analysis been performed appropriately and rigorously? 

Reviewer #1: Yes

Reviewer #2: No

4. Have the authors made all data underlying the findings in their manuscript fully available?

Reviewer #1: Yes

Reviewer #2: Yes

5. Is the manuscript presented in an intelligible fashion and written in standard English?

Reviewer #1: Yes

Reviewer #2: Yes

6. Review Comments to the Author

Reviewer #1: Thank you for your efforts in reviewing this paper. My questions, suggestions, and comments are satisfactorily addressed. The paper is acceptable.

Reviewer #2: General Comments

Overall, the manuscript still currently does not describe how their analysis is novel and different from previous work. There are many papers on deep convection transporting pollution to the upper troposphere over Asia (eg., https://www.science.org/doi/full/10.1126/science.1182274,
https://agupubs.onlinelibrary.wiley.com/doi/full/10.1029/2007GL030638 and many others). The paper currently reads that the presented analysis is the ‘first’ to show this, which is incorrect. Much more discussion of previous studies and how this work is novel is needed.

The authors must better describe why they only look at data from 2012-2013. There is much more data than this available from IAGOS https://amt.copernicus.org/articles/18/129/2025/. Why compare their data to previous NASA aircraft missions, but not other years of IAGOS data? The authors also need to better describe how their work is different from other papers on IAGOS CO trends that cover longer time periods. See the following (non-comprehensive) list of papers:

1. https://acp.copernicus.org/articles/24/13975/2024/acp-24-13975-2024.html Also discusses elevated CO over East Asia.

2. https://acp.copernicus.org/articles/18/5415/2018/ Also discusses CO seasonality and elevated CO over East Asia.

3. https://acp.copernicus.org/articles/23/14039/2023/acp-23-14039-2023.pdf Also discusses CO seasonality and highlights the impact of anthropogenic emissions.

Also, there was no marked up manuscript submitted as part of this revision, which made the changes difficult to assess.

Specific Comments

Line 45 – I do not know what the authors mean by “the first time”. What about the MOZAIC flights for example mentioned in Section 2.2? Did none of the MOZAIC flights go over the Pacific? Is this really necessary to state at all? You even say this on line 82 “Two decades of MOZAIC measurements have affirmed the reliability of this instrument for continuous operation over 6-12 months aboard commercial passenger aircraft.”

Line 172 – Again are these really the first?

Figure 5f – Why are there blue box and whiskers?

Figure 6 – Why are these regions not identical to the regions in Figure 5? Example, Fig. 5f is 160-110W, 75-25S but Fig. 6e is 160-110W, 70-20N. I don’t think the altitude trace is needed, just say the measurements are between 9-13

Line 241 – I am again not convinced “first time” is really appropriate here.

Section 3.2.4 – This should read ‘Comparisons With’

Figure 7 – I would strongly suggest plotting similar time periods and locations on top of each other. You could plot for example the box whisker from Fig. 7l and 7d on the same plot to better visualize differences.

Line 283 – The suggestion above for Fig. 7 would make this point stronger.

Figure 13 – What further insight does this provide that hasn’t already been shown in Fig. 5-7?

7. PLOS authors have the option to publish the peer review history of their article (what does this mean?). If published, this will include your full peer review and any attached files.

Reviewer #1: No

Reviewer #2: No

---

## [Author Response · Author response to Decision Letter 2]

13 Mar 2025

Point-by-Point Replies to Reviewer 2

We sincerely thank the reviewer for the very insightful comments, which have significantly increased the clarity and overall quality of our revised manuscript.

We have read reviewer’s comments carefully and revise our manuscript accordingly. We have paid specific attention to the request to highlight the differences in our study vs other studies in the past. We have put our work into the context of previous scientific perspective. We have explained how our work differs/confirms the previous studies cited by the reviewer.

Below, please find our detailed, point-by-point responses.

General Comments

Reviewer Comment:

Overall, the manuscript still currently does not describe how their analysis is novel and different from previous work.

Revised Reply:

The detailed analysis of IAGOS Pacific flights and the spatial distribution of CO measurements over the open ocean of the North Pacific upper troposphere, between 120∘E (eastern Eurasia continent and the beginning of the North Pacific atmosphere) and 120∘W (western edge of North America, end of the North Pacific atmosphere), during an 8‐month period (July 2012 to February 2013) has not been previously reported. These measurements taken during this 8-month period over such a wide-spread of the North Pacific upper troposphere can only be made possible by the long-hull flights using A340-300 airplane to across North Pacific. Hence, these in-situ flight measurements are rare to characterize long-range transport of CO from Asia

to the downwind of the North Pacific upper troposphere.

Barret et al. (2025) used IAGOS data to validate satellite (IASI-A, IASI-B, and IASI-C; Infrared Atmospheric Sounding Interferometer) CO measurements, covering the period 2008 to 2019. They presented results from Taipei (25.09∘N, 121.24∘E) and Nagoya (34.85∘N,136.81∘E), located over the Northwestern Pacific, to verify the IAGOS measurements. Barret et al. (2025) showed time-series results of CO in the troposphere between the surface–600 hPa layer and the 600 hPa–200 hPa layer over Taipei City for two periods: July 2012–February 2013 and 2015–mid 2018. Additional data were available from container measurements made by CARIBIC.

Lebourgeois et al. (2024) used IAGOS measurements from 2002 to 2019, the FLEXPART trajectory model, and surface CO emissions inventories to attribute the ground-level sources of elevated CO measured by IAGOS on a global scale. Elevated CO was linked to surface emissions (anthropogenic, biomass burning, or a combination of both) by analyzing the 20-day backtrajectory from each elevated CO measurement. In the East Asia upper troposphere—defined as 90∘E to 150∘E and 15∘N to 45∘N—elevated CO levels (175 ppb) occurred during the summer months, compared to 130 ppb in winter. During summer (June, July, August), about 95% of elevated CO originated from anthropogenic emissions, with the remainder from biomass burning and mixed sources. In winter (December, January, February), approximately 90% of elevated CO was attributed to anthropogenic sources, with the remainder from biomass burning and mixed emissions.

Cohen et al. (2018) discussed the climatology and long-term evolution of ozone and carbon monoxide in the upper troposphere–lower stratosphere (UTLS) at northern midlatitudes, as observed by IAGOS from 1995 to 2013 on a global scale. In the East Asia region, defined as 105∘E–145∘E and 30∘N–50∘N, elevated CO levels were found in the upper troposphere during late spring (May) and summer (June and July), while the lower stratosphere exhibited elevated CO levels during the summer (June, July, August). Cohen et al. (2018) also reported negative CO trends (5th percentile, mean value, and 95th percentile) over the 105∘E–145∘E and 30∘N–N50∘N area of the East Asia upper troposphere. It is noteworthy that IAGOS Pacific measurements commenced in July 2012 using China Airlines’ in-service A340-300 aircraft.

Tsivlidou et al. (2023) employed FLEXPART 20-day backtrajectories based on elevated CO measured by IAGOS over China and the tropical West Pacific (25∘N–25∘S) during 2002–2020 to attribute the sources of elevated CO. The methodology used was similar to that of Lebourgeois et al. (2024). Again, IAGOS Pacific measurements were initiated in July 2012.

The aforementioned studies utilized extensive data periods (2008–2019, 2002–2019, 1995–2013, 2002–2020) from IAGOS measurements, covering areas primarily over the Atlantic, Western Europe, North America, South America, and Africa, as well as some flights from Europe to Japan, China, and Korea. In contrast, the IAGOS Pacific measurements, started in July 2012, were conducted over the open ocean of the North Pacific (120∘E–120∘W) for flights from East Asia to North America.

Previous studies using IAGOS data have provided a broad, seasonal (summer, winter) view of CO characteristics over the eastern edge of the Eurasian continent and at the intersection of Eurasia and the northwestern Pacific. In this work, we complement these long-term and seasonal (summer, winter) studies by providing a detailed, zoomed-in analysis on a day-to-day and month-to-month basis for the first 8 months of IAGOS Pacific flights across the open ocean of the North Pacific upper troposphere. This detailed analysis captures episodic events that are missed in seasonal data and reveals transient CO peaks critical for understanding episodic pollution transport. We focus on July 2012–February 2013, marking the initiation of IAGOS Pacific flights, which provides a baseline for subsequent trends, supported by a sufficient data density for detailed analysis. Future work may explore later years to assess temporal evolution.

We present the distribution of elevated CO over the North Pacific open ocean upper troposphere and conduct backtrajectory calculations to identify the sources of both elevated and low CO. For the lower troposphere, we analyze a total of 520 CO profiles over Taipei (at a vertical resolution of 50 m), performing profile-by-profile backtrajectory analysis for both elevated and low CO measurements. The comparisons reveal distinct sources of air masses arriving in Taipei, complementing the upper tropospheric focus on sources of elevated versus low CO.

Reviewer Comment:

There are many papers on deep convection transporting pollution to the upper troposphere over Asia

Revised Reply:

We have included the following description in the introduction:

Asian anthropogenic emissions and boreal biomass burning dominate CO in the North Pacific upper troposphere (Jaffe et al., 1999). Fast uplift via convection and mid-latitude cyclones facilitates CO injection into the upper troposphere (Jiang et al., 2007; Randel et al., 2010). Spring and early summer show heightened CO due to efficient vertical transport and favorable wind patterns (e.g., the Westerlies). Satellite observations (Jiang et al., 2007), in situ aircraft observations (Cohen et al., 2018; Lebourgeois et al., 2024; Barret et al., 2025), and models such as HYSPLIT (Chen et al., 2021), FLEXPART (Tsivlidou et al., 2023), and GEOS-Chem (Liang et al., 2004; Miyazaki et al., 2020) are frequently employed to study the long-range transport of air pollutants from Asian anthropogenic and biomass burning emissions. While Jiang et al. (2007) used satellite data to demonstrate seasonal CO transport, our IAGOS profiles provide in situ validation with finer temporal detail over the North Pacific and allow refined source attribution via backtrajectory analysis.

Reviewer Comment (related to citation examples):

e.g., {https://www.science.org/doi/full/10.1126/science.1182274}

Revised Reply:

Randel et al. (2010) used the Atmospheric Chemistry Experiment Fourier Transform Spectrometer (ACE-FTS) and MLS satellite measurements of atmospheric hydrogen cyanide (HCN) along with the three-dimensional climate model WACCM3 to study elevated HCN in the lower stratosphere during 2004–2009 over regions impacted by the Asian summer monsoon. Their results demonstrate that surface air enriched in HCN is rapidly transported into the stratosphere by deep convection. They also noted that, in addition to the maximum vertical transport during boreal summer (June–August), seasonally varying HCN sources include biomass burning over Indonesia and Africa during boreal spring (March–May) and over Africa and South America during austral spring (September–November), with deep convection facilitating the transport of these emissions into the upper troposphere.

Reviewer Comment:

The paper currently reads that the presented analysis is the ‘first’ to show this, which is incorrect. Much more discussion of previous studies and how this work is novel is needed.

Revised Reply:

We have removed all descriptive words such as “first” from the revised manuscript. Our work is positioned as an extension of previous missions and studies. We have incorporated the references provided by the reviewer and clarified the novelty of our approach by emphasizing the high temporal resolution (day-to-day and month-to-month) analysis of IAGOS Pacific flights during the initial 8-month period, which captures transient CO peaks that longer-term seasonal studies may overlook.

Reviewer Comment:

The authors must better describe why they only look at data from 2012-2013. There is much more data than this available from IAGOS.

Revised Reply:

Barret et al. (2025) used IAGOS data to validate satellite (IASI-A, IASI-B, and IASI-C; Infrared Atmospheric Sounding Interferometer) CO data, covering 2008–2019. They presented time-series results of CO in the troposphere for two periods: July 2012–February 2013 over Taipei and during 2015–mid 2018. Although longer datasets exist, we focused on July 2012–February 2013 because it marks the initiation of IAGOS Pacific flights using a China Airlines Airbus A340-300. This period offers sufficient data density for a detailed, high-temporal resolution analysis and serves as an important baseline for subsequent trend analysis. Future work could extend this approach to additional years to assess temporal evolution.

Reviewer Comment:

Why compare their data to previous NASA aircraft missions, but not other years of IAGOS data? The authors also need to better describe how their work is different from other papers on IAGOS CO trends that cover longer time periods.

Revised Reply:

1. Comparison with Previous Studies:

Lebourgeois et al. (2024) used IAGOS 2002–2019 measurements, the FLEXPART trajectory model, and surface CO emissions inventories to attribute sources of elevated CO on a global scale. They found that in the East Asia upper troposphere (defined between $90^\circ$E and $150^\circ$E and $15^\circ$N–$45^\circ$N), elevated CO concentrations (175 ppb) occur in the summer months compared to 130 ppb in winter, with anthropogenic emissions dominating during summer and biomass burning contributing more during winter.

Cohen et al. (2018) analyzed the long-term evolution of ozone and carbon monoxide in the upper troposphere–lower stratosphere from 1995 to 2013 and identified seasonal trends in the East Asia region ($105^\circ$E–$145^\circ$E and $30^\circ$N–$50^\circ$N). They reported negative trends in CO.

Tsivlidou et al. (2023) applied FLEXPART backtrajectory analysis to elevated CO measured by IAGOS over China and the tropical western Pacific during 2002–2020. Their method is similar to that used by Lebourgeois et al. (2024).

2. Our Work’s Novelty:

In contrast to these studies that provide a zoomed-out view on seasonal or decadal trends, our work provides a detailed, high-temporal resolution (day-to-day and month-to-month) analysis of the first 8 months (July 2012–February 2013) of IAGOS Pacific flights. This focused period allows us to capture episodic events and transient CO peaks that are often averaged out in longer-term studies. By doing so, we can better understand the mechanisms of episodic pollution transport and provide a baseline for future trend analyses.

Reviewer Comment:

Also, there was no marked-up manuscript submitted as part of this revision, which made the changes difficult to assess.

Revised Reply:

We have provided a marked-up version of the revised manuscript for your convenience.

Specific Comments

Reviewer Comment (Line 45):

I do not know what the authors mean by “the first time”. What about the MOZAIC flights, for example, mentioned in Section 2.2? Did none of the MOZAIC flights go over the Pacific? Is this really necessary to state at all? You even say this on line 82 “Two decades of MOZAIC measurements have affirmed the reliability of this instrument for continuous operation over 6-12 months aboard commercial passenger aircraft.”

Revised Reply:

We have revised the wording and removed the term “first” from the revised manuscript to avoid any ambiguity. Our focus is on the specific characteristics of the IAGOS Pacific flights, not on asserting uniqueness relative to MOZAIC flights.

Reviewer Comment (Line 172):

Again, are these really the first?

Revised Reply:

We have removed the word “first” from the revised manuscript to accurately represent our findings without implying novelty that is not supported by the literature.

Reviewer Comment (Figure 5f):

Why are there blue box and whiskers?

Revised Reply:

We have revised the color scheme for Figure 5f to ensure consistency with the other plots. The updated figure now uses a uniform color palette that improves clarity and comparability across figures.

Reviewer Comment (Figure 6):

Why are these regions not identical to the regions in Figure 5? For example, Fig. 5f is 160–110°W, 75–25°S but Fig. 6e is 160–110°W, 70–20°N. I don’t think the altitude trace is needed, just say the measurements are between 9–13.

Revised Reply:

Figure 5 displays measurements from all flights made by the IAGOS Package 1 on the CAL A340-300 B18316, whereas Figure 6 focuses on measurements over the Northern Hemisphere midlatitudes, particularly those from flights across the North Pacific. The altitude traces in Figure 6 indicate the data density per flight. We have now clarified this point in the revised manuscript by stating, “Figure 6 isolates Northern Hemisphere midlatitude data (e.g., $20^\circ$N–$70^\circ$N) to highlight North Pacific trends, with altitude traces retained to reflect data density per flight,” and we have removed unnecessary altitude markers.

Reviewer Comment (Line 241):

I am again not convinced “first time” is really appropriate here.

Revised Reply:

We have removed the word “first” from the revised manuscript to avoid any potential misinterpretation regarding the novelty of our observations.

Reviewer Comment (Section 3.2.4):

This should read ‘Comparisons With’

Revised Reply:

We have corrected the heading in Section 3.2.4 to read “Comparisons With” as suggested.

Reviewer Comment (Figure 7):

I would strongly suggest plotting similar time periods and locations on top of each other. You could plot, for example, the box-whisker from Fig. 7l and 7d on the same plot to better visualize differences.

Revised Reply:

We have rearranged the plots so that those representing similar time periods (by month) are placed adjacent to each other. For example, Figure 7d now shows measurements during September–October 2012 while Figure 7l shows measurements during February–April 2013. We have used identical x-axis ranges and added a highlighted dashed vertical line to facilitate direct comparisons between the figures.

Reviewer Comment (Line 283):

The suggestion above for Fig. 7 would make this point stronger.

Revised Reply:

We have added a vertical black dashed line at 200 ppbv in Figure 7 to underscore the shift in CO concentrations: in 1991, most values ranged from 50 to 150 ppbv (Figure 7b), w

---

## [Decision Letter · Decision Letter 2]

9 Apr 2025

Elevated Carbon Monoxide Observations Over the North Pacific Upper Troposphere (July 2012 - February 2013)

PONE-D-24-32569R2

Dear Dr. Wang,

We’re pleased to inform you that your manuscript has been judged scientifically suitable for publication and will be formally accepted for publication once it meets all outstanding technical requirements.

The second reviewer thinks that the manuscript still lacks novelty. However, in PLOS ONE evaluation of research is done on the basis of scientific validity, strong methodology, and high ethical standards—not perceived novelty as such. Therefore I have decided to accept your manuscript for publication.

Kind regards,

Markus Metsälä

Academic Editor

PLOS ONE

Additional Editor Comments (optional):

Reviewers' comments:

Reviewer's Responses to Questions

**Comments to the Author**

1. If the authors have adequately addressed your comments raised in a previous round of review and you feel that this manuscript is now acceptable for publication, you may indicate that here to bypass the “Comments to the Author” section, enter your conflict of interest statement in the “Confidential to Editor” section, and submit your "Accept" recommendation.

Reviewer #2: (No Response)

2. Is the manuscript technically sound, and do the data support the conclusions?

Reviewer #2: No

3. Has the statistical analysis been performed appropriately and rigorously? 

Reviewer #2: No

4. Have the authors made all data underlying the findings in their manuscript fully available?

Reviewer #2: Yes

5. Is the manuscript presented in an intelligible fashion and written in standard English?

Reviewer #2: Yes

6. Review Comments to the Author

Reviewer #2: The authors have addressed the comment about discussing previous work by providing a literature list of previous IAGOS papers. They cite one paper, Lebourgeois et al., 2024, that attributed elevated CO measured by IAGOS primarily to anthropogenic emissions. The main difference in the work here is the use of IAGOS measurements over the North Pacific from East Asia to North America. The authors say that their ‘detailed analysis captures episodic events that are missed in seasonal data and reveals transient CO peaks critical for understanding episodic pollution transport.’ However, this work does not provide any additional insight into understanding episodic pollution transport other than showing that convection lifts anthropogenic pollution into the upper troposphere which is not a new finding.

If the authors would like to publish the work as is, I would suggest they submit a measurement report, for example to Atmospheric Chemistry and Physics https://www.atmospheric-chemistry-and-physics.net/about/manuscript_types.html. If the authors would like to increase the scientific value of this work, more analysis is need. In the introduction, the authors state that they will present “continuous in-situ measurements of CO and ozone (O3) conducted during routine commercial flights” but they do not show ozone in the subsequent analysis. The authors could use their data to evaluate model transport of pollution for example, which would be valuable. Or they could look at changes in the ozone to CO ratio indicative of ozone production or destruction during transport. The authors say for example on page 6, “These in-situ CO measurements are crucial for validating model transport processes and satellite remote sensing of CO in the troposphere.” That is true, and that would be a valuable use of these measurements. The current finding that upper tropospheric CO is due to anthropogenic emissions in Asia is not sufficiently novel. This statement on page 10, line 344 is not clearly supported by a figure “Regions of the Pacific atmosphere immediately downwind of the Asian continent (between 120◦E and 150◦E) exhibit higher CO concentrations compared to areas further downwind (east of 150◦E).” Are we supposed to somehow infer this from Fig. 7? This is only qualitatively discussed in the subsequent paragraph. A quantitative assessment of trends would also be a valuable scientific contribution. As it stands, I think this work would be a fine measurement report but does not achieve the level of analysis for a scientific research paper.

7. PLOS authors have the option to publish the peer review history of their article (what does this mean?). If published, this will include your full peer review and any attached files.

Reviewer #2: No

---

## [Editor Report · Acceptance letter]

PONE-D-24-32569R2

PLOS ONE

Dear Dr. Wang,

I'm pleased to inform you that your manuscript has been deemed suitable for publication in PLOS ONE. Congratulations! Your manuscript is now being handed over to our production team.

Kind regards,

on behalf of

Dr. Markus Metsälä

Academic Editor

PLOS ONE